

# 1 Spatiotemporal variation of snow depth in the Northern
# 2 Hemisphere from 1992 to 2016

Xiongxin Xiao[1, 2], Tingjun Zhang[1, 4], Xinyue Zhong[3], Xiaodong Li[1], Yuxing Li[1]
[1]Key Laboratory of Western China's Environmental Systems (Ministry of Education), College of Earth
and Environment Sciences, Lanzhou University, Lanzhou 730000, China
[2]School of Remote Sensing and Information Engineering, Wuhan University, Wuhan 430079, China
[3]Key Laboratory of Remote Sensing of Gansu Province, Cold and Arid Regions Environmental and
Engineering Research Institute, Chinese Academy of Sciences, Lanzhou 730000, China
[4]University Corporation for Polar Research, Beijing 100875, China.
*Correspondence to*: Tingjun Zhang (tjzhang@lzu.edu.cn)
**Abstract:** Snow cover is an effective best indicator of climate change due to its effect on regional and
global surface energy, water balance, hydrology, climate, and ecosystem function. We developed a long
term Northern Hemisphere daily snow depth and snow water equivalent product (NHSnow) by the
application of the support vector regression (SVR) snow depth retrieval algorithm to historical passive
microwave sensors from 1992 to 2016. The accuracies of the snow depth product were evaluated
against observed snow depth at meteorological stations along with the other two snow cover products
(GlobSnow and ERA-Interim/Land) across the Northern Hemisphere. The evaluation results showed
that NHSnow performs generally well with relatively high accuracy. Further analysis were performed
across the Northern Hemisphere during 1992-2016, which used snow depth, total snow water
equivalent (snow mass) and, snow cover days as indexes. Analysis showed the total snow water
equivalent has a significant declining trends (~5794 km$^3$ yr.$^{-1}$, 12.5% reduction). Although spatial
variation pattern of snow depth and snow cover days exhibited slight regional differences, it generally
reveals a decreasing trend over most of the Northern Hemisphere. Our work provides evidence that
rapid changes in snow depth and total snow water equivalent are occurring beginning at the turn of the
21st century with dramatic, surface-based warming.

## 26  1. Introduction

Seasonal snow cover is an important component of the climate system and global water cycle that
stores large amounts of freshwater and play major impacts on the surface energy budget, climatology
and water management (Immerzeel et al., 2010;Zhang, 2005;Robinson and Frei, 2000;Tedesco et al.,
2014). On account of the high albedo and low heat conductivity properties of snow, snow cover may
directly modulate the land surface energy balance (Flanner et al., 2011), influence on soil thermal
regime (Zhang et al., 1996;Zhang, 2005), and indirectly affect atmospheric circulation (Cohen et al.,
2012;Zhang et al., 2004;Li et al., 2018). Most jurisdictions in the Northern Hemisphere rely on natural
water storage provided by snowpack (Diffenbaugh et al., 2013;Barnett et al., 2005), supplying water for
domestic and industrial use (Sturm, 2015;Qin et al., 2006). Accurate estimation of and reliable
information on snow cover spatial and temporal change at regional and global scales is very critical for
climate change monitoring, model evaluation and water source management (Brown and Frei,
2007;Flanner et al., 2011).
Snow depth (SD) is most commonly measured using in situ observations. Given the sparseness of
measurements, it is not possible to fully capture spatial variability of snow cover. Although the in situ
observation method is accurate, it is unrealistic in mountain regions and low population zones because
it is labor, material and financial resource intensive. Remote sensing is the most effective and powerful
way of obtaining information of snow cover over larger areas (Foster et al., 2011). Optical remote
sensing is capable of observing large areas of snow; however, it is unable to observe the Earth's surface
under cloudy conditions (Foster et al., 2011;Che et al., 2016;Dai et al., 2017). However, microwave
remote sensing has this potential and is an attractive alternative to optical remote sensing under all
weather conditions and round the clock. It can also be used to estimate SD and snow water equivalent
(SWE) due to the interaction with snowpack by providing dual polarization data at different
frequencies (Chang et al., 1987;Che et al., 2008;Takala et al., 2011).
Snow cover products derived from passive microwave (PM) data have been widely applied to
investigate regional and global climate change, and validate hydrological and climate models (Brown et
al., 2010;Brown and Robinson, 2011;Dai et al., 2017). Progress in satellite data acquisition, as well as
SD/SWE retrieval algorithm development, have led to a global improvement in snow monitoring (Qin
et al., 2006;Snauffer et al., 2016). The PM brightness temperature of the SMMR (Scanning
Multichannel Microwave Radiometer), SSM/I (Special Sensor Microwave Imager), AMSR-E
(Advanced Microwave Scanning Radiometer for Earth Observing System), AMSR2 (Advanced
Microwave Scanning Radiometer 2 on the Global Change Observation Mission – Water), SSMIS
(Special Sensor Microwave Imager), SSM/I (Special Sensor Microwave Imager Sounder) and,
FY-3B/C (Fengyun-3 satellite B/C) are available and several algorithms have been developed to





estimate SD and SWE using PM brightness temperature data (Chang et al., 1987;Dai et al., 2012;Xiao
et al., 2018;Pulliainen, 2006;Takala et al., 2011;Che et al., 2008;Foster et al., 1997).
Most retrieval algorithms operate on the principle that the difference in brightness temperature
between 18 and 37 GHz reflects the quantity of SD and SWE (Chang et al., 1987). Over and
underestimated trends are prevalent in these linear SD and SWE retrieval algorithms (Gan et al., 2013)
for which there are two possible and reasonable explanations. One is that vegetation overlaying snow
attenuates its microwave scatter signal and results in underestimating SD and SWE from PM data (Che
et al., 2016;Vander Jagt et al., 2013). To reduce the effect of tree canopy, a forest fraction was
introduced into retrieval algorithm developed to estimate SD and SWE (Foster et al., 1997;Che et al.,
2008), or the retrieval algorithm was constructed based on particular land cover types (Goïta et al.,
2003;Che et al., 2016;Derksen et al., 2005;Foster et al., 2009). The other explanation is that the
relationship between snow properties (SD or SWE) and the PM brightness temperature is non-linear.
Newer approaches (e.g. artificial neural networks, support vector regression, decision tree) have
emerged using data-mining and have been explored to retrieve SD and SWE that are intended to
replace traditional linear methods (Gharaei-Manesh et al., 2016;Tedesco et al., 2004;Liang et al.,
2015;Forman et al., 2013;Xue and Forman, 2015). However, there are remain some limitations for
these retrieval algorithms due to the diversity of land cover types and the spatiotemporal heterogeneity
of snow physical properties.
Numerous studies have reported the changes in snow cover extent (SCE) at regional and
hemispheric scales (Rupp et al., 2013;Dai et al., 2017;Derksen and Brown, 2012;Brown and Robinson,
2011;Huang et al., 2016). Huang et al. (2017) repored the impact of climate and elevaion on snow
cover varition in Tibetan Plateau, including SWE, snow cover area and, snow cover days. Hori et al.
(2017) developed a 38-year Northern Hemisphere daily snow cover extent product and analyzed
seasonal Northern Heimsphere snow cover extent variation trends. In this study, SD was selected as
basis for analyzing spatiotemporal change of snow cover. SD provides an additional dimension to snow
cover characteristics. Barrett et al. (2015) explored intra-seasonal variability in springtime Northern
Hemisphere daily SD change by the phase of the Madden–Julian oscillation. Wegmann et al. (2017)
compared four long-term reanalysis datasets with Russian SD observation data. However, this study
only focused on snowfall season (October and November) and snowmelt season (April). SD change
trends have also been analyzed at regional scales (Ye et al., 1998;Dyer and Mote, 2006). Several studies



quantified the spatial and temporal changes consistency of SWE or snow mass derived from satellite
data (Mudryk et al., 2015) but these studies have focused on the limited dimension of snow cover
variation. Dyer and Mote used a gridded dataset to study regional and temporal variability of SD trends
across North America from 1960-2000 (Dyer and Mote, 2006) and the characteristic of seasonal snow
extent and snow mass in South America form 1979 to 2006 was descripted and reported (Foster et al.,

6     2009).

7       There are, however, very limited data (station data, satellite data or otherwise) that can provide

both SD and SWE on a hemispheric scale. This paper describes the approach to develop a consistent
25-year of daily SD and SWE of Northern Hemisphere utilized multi-source data. The primary
objective of this study is to develop 25 years (1992-2016) hemispherical SD and SWE product
(hereafter referred to as the NHSnow) with a 25 km spatial resolution using SVR SD retrieval
algorithm. This paper will address the following questions: 1) How consistent are NHSnow and other
sourced snow cover datasets with the in situ SD observation? 2) What is the spatiotemporal variability
of snow cover in the Northern Hemisphere from 1992-2016? Meanwhile, it is extremely challenging to
make extensive quantitative validation of SD and SWE estimates.

16       This paper is organized in five sections, as follows. Section 2 describes the data sets used in this

study. The methods of data preprocessing and snow cover products generation were provided in
Section 3. Next, we describe NHSnow validation against in-situ snow observation record, exhibit the
variability of snow cover in the Northern Hemisphere and discuss the potential effect factors for the
variation results utilized NHSnow data (Section 4). Finally, section 5 summarizes the work of this
paper.
**2 Datasets**
**2.1 Passive microwave data**

24       Because cloud often appear in the snow cover region or condition, during the winter season often

conceals snowfall possibility, here is particularly advantageous using passive microwave remote
sensing. SSM/I and SSMIS is PM radiometer onboard United States Defense Meteorological Satellites
Program (DMSP) satellite (available from the National Snow and Ice Data Center,
http://nsidc.org/data/NSIDC-0032). The SSM/I (F11 and F13) dataset from this platform, as well as



SSMIS (F17), present with the equal-area scale earth grid (EASE-Grid) format and 25 km spatial
resolution (Brodzik and Knowles, 2002;Armstrong, 2008;Wentz, 2013;Armstrong and Brodzik, 1995)
(Table 1). The snow cover area and SD derived from SSM/I (F11) and SSM/I (F13) data have high
consistency rendering the calibration between these two sensors for snow cover area and SD
unnecessary (Dai et al., 2015). To minimize the melt-water effect to some extent, which can change the
microwave emissivity of snow, only descending orbit (nighttime) passive microwave data were used
(Foster et al., 2009).
**2.2 Ground-based data**

9       Ground SD observation are used to construct and verify the SD retrieval model in this study from

two sources of daily SD observation. The first is the Global Surface Summary of the Day (GSOD)
dataset provided by National Oceanic and Atmospheric Administration (NOAA)
(https://data.noaa.gov/dataset/dataset/global-surface-summary-of-the-day-gsod). This online dataset,
which began in 1929, is derived from the Integrated Surface Hourly (ISH) dataset (Xu et al., 2016).
There are fourteen daily elements in GSOD dataset, including SD measured at 0.1 inch. The missing of
SD or reported 0 on the day would be marked 999.9. Data at approximately 30000 meteorological
stations were recorded of which 9000 typically are valid. In our study period and area, more than 17
000 meteorological station were selected with records from 1991 and a location far from large water
bodies.

19       To supplement data from stations that were not reporting during the study periods, ground-based

measurements of daily SD were gathered from an additional 635 Chinese meteorological stations
available at the National Meteorological Information of China Meteorological Administration (Xiao et
al., 2018;Zhong, 2014). These daily SD records begun in 1957 include SD (unit, cm), observation time,
and geographical location information available (http://data.cma.cn/en).
**2.3 Topographic and land cover data**

25       We also used topography as an auxiliary information to estimate SD (Xiao et al., 2018). Elevation

was available from ETOPO1 at a resolution of 1 arc-minute (Amante, 2009) available at
(http://www.ngdc.noaa.gov/mgg/global/). To match the resolution of the PM brightness temperature
data with 25 km spatial resolution, we resampled the ETOPO1 to 25 km resolution (Fig. 1).



To increase the accuracy of SD estimates for different land cover types, we both used MODIS land
cover (MCD12Q1 V051) from 2001 to 2013 (Friedl and Sulla-Menashe, 2011;Friedl et al., 2010) and
Advanced Very High Resolution Radiometer (AVHRR) Global Land Cover classification generated by
the University of Maryland Department of Geography. The MCD12Q1 International
Geosphere-Biosphere Program (IGBP) classification scheme divides land surface into 17 types, which
were reclassified into five classes according to Xiao et al (2018) study.
AVHRR imagery was acquired between 1981-1994 from the NOAA-15 satellite (Hansen et al.,
2000) and were categorized into fourteen land cover classes at 1 km resolution. These data allowed us
to adjust the proposed snow-depth retrieval algorithm by reclassifying the fourteen native land cover
classes into five classes (water, forest, shrub, prairie and, bare-land) at 25 km spatial resolution (Table
A.). MCD12Q1 is available at site https://earthdata.nasa.gov/, while AVHRR land cover data is
available from http://www.landcover.org/data/landcover/.
**2.4 Satellite snow cover datasets**
Two kinds of snow cover datasets were utilized based on two criteria: covering the Northern
Hemisphere and long-term availability. We selected GlobSnow and ERA-Interim/Land which are
widely used in global and regional climate change studies (Snauffer et al., 2016;Hancock et al.,
2013;Mudryk et al., 2015). These datasets were used to compare with the NHSnow SD product.
In November 2013, the European Space Agency (ESA) released the GlobSnow Version 2.0 SWE
and Snow Extent (SE) data for the Northern Hemisphere (Takala et al., 2011;Pulliainen, 2006). These
data include all non-mountainous areas in the Northern Hemisphere and are available online
(http://www.globsnow.info/). Processing includes data assimilation based on combining satellite PM
remote sensing data (SMMR, SSM/I and SSMIS), spanning December 1979 to May 2016, with
ground-based observation data in a data assimilation scheme to derive SWE. GlobSnow Version 2.0
(hereinafter referred as GlobSnow) provides three kinds of temporal aggregation level products with
25 km spatial resolution: daily, weekly and monthly. This dataset covers all land surface areas in a
band between 35° N ~ 85° N excluding mountainous regions, glaciers and Greenland. To convert
between SD and SWE using GlobSnow, the snow density is held constant at 0.24 g/cm$^3$ (Sturm et al.,
2010;Hancock et al., 2013;Che et al., 2016).
ERA-Interim/Land (Balsamo et al., 2015) is a global land-surface reanalysis product with data



from January 1979 to December 2010 based on ERA-Interim meteorological forcing. It is produced by
a land-surface model simulation using the Hydrology Tiled ECMWF Scheme of Surface Exchange
over Land (HTESSEL), with meteorological forcing from ERA-Interim. Dutra et al. (2010) described
the snow scheme and demonstrated the verification using field experiments. "SD", which actually is
SWE, is one of the thirteen parameters provided. We should convert SWE to SD using the associated
snow density data. These two datasets are available online
(http://apps.ecmwf.int/datasets/data/interim-land/type=an/). To maximum the proximity to the
descending orbit time of passive microwave sensor, the data with analysis type at 6 o'clock were used
in this study, and the spatial resolution of these data is 0.125 degree.
**2.5 Snow classification data**
In order to accurately estimate SWE, snow classification data were used to convert SD into SWE.
Global Seasonal Snow Classification System was defined by Sturm et al. (1995) based on snow
physical properties (SD, thermal conductivity, snow density snow layers, degree of wetting, etc.), and
seasonal snow cover. Snow cover were categorized into six snow classes (tundra, taiga, alpine,
maritime, prairie, and ephemeral) plus water and ice fields (Figure 2). Snow classification data can be
accessed from the National Center for Atmospheric Research (NCAR)/Earth Observing Laboratory
(EOL) (https://data.eol.ucar.edu/dataset/6808). The snow classification dataset was developed and
tested for the Northern Hemisphere at 0.5-degree spatial resolution(Sturm et al., 1995).
**3 Methods**
**3.1 Theoretical basis**
Snow distribution is affected by various factors, but not limited to, vegetation (Che et al.,
2016;Vander Jagt et al., 2013), soil and air temperature (Forman and Reichle, 2015;Grippa et al.,
2004;Dai et al., 2017), topography and wind (Smith and Bookhagen, 2016). The snow retrieval process
uses DS and other parameters (A, T, G, L, D ...) to yield snow parameters (e.g. SD, Eq. 1) (Xiao et al.,

25  2018).

$$[S] = g\,(A, T, G, L, DS, D ...) + \varepsilon \tag{1}$$

where g ( · ) denotes the retrieval function. DS is the digital signal from remote sensing sensor (PM,





active microwave, visible spectral remote sensing etc.), A is the atmosphere (wind speed, air
temperature, humidity, precipitation etc.), T is the topography (latitude, longitude, elevation, terrain
slope, aspect etc.), L is the location (latitude, longitude), G is the ground (ground surface temperature,
vegetation type etc.), S is the snow properties (snow grain size, density, reflectance, SD, SWE etc.), D
is the day of year and ε is the residual error or uncertainty that describes the relationship between
sensor signal and measured snow properties.
The SVR SD retrieval algorithm also follows the snow retrieval process (Eq. 1). We utilized ten
parameters were as input parameters, including PM brightness temperature (19 GHz, 37 GHz, 85 GHz,
or 91 GHz) with vertical and horizontal polarizations, geophysical location (latitude and longitude),
elevation and, the measured SD. The output parameter is the estimated SD. Apart from above factors,
the SVR SD retrieval algorithm also considers other influence factors, including wet snow, land cover
types and day of year (Xiao et al., 2018) to improve the accuracy of estimated SD. Day of year have
been converted into three snow cover stages, which mean indirectly considering snow properties
evolution.
**3.2 Processing flow overview**
The SVR SD retrieval algorithm first proposed by Xiao et al. (2018), which indirectly considers
seasonal variation and vegetation influence in the evolution of snow properties, was used to estimate
SD. In Eurasia, it was found that the SVR SD retrieval algorithm performs much superior with reduced
uncertainties compared based upon the correlation coefficient (R), mean absolute error (MAE), and
root mean squared error in Xiao et al. (2018) study. It should be noted that this study used daily
observation in the Northern Hemisphere with exception of July and August. Here, we provide more
detailed but different descriptions for the SVR SD retrieval algorithm in several steps (Fig. 3). The
detailed descriptions of the other steps can refer to the Xiao et al paper (Xiao et al., 2018) not repeated
here.
Step 3. Due to our study period pre-dates MODIS data, we used AVHRR land cover as suppliment
data. MODIS and AVHRR land cover were reclassified into four classes (forest, prairie, shrub and
bare-land) which were bases of construting SD retrieval sub-model. Table A (in appendix) describes the
reclassification scheme of AVHRR land cover is described. MODIS land cover reclassification schemes
were documented in Xiao et al. (2018). Because of the relative stability of land cover change, MODIS


land cover in 2013 was used for each year during 2013–2016. Similarly, MODIS land cover in 2001
was used in each year during 1998–2001, and AVHRR land cover data were used for 6 years
(1992–1997).

4        Step 6.1 Construction of a subcontinental model. It needs to be stressed that the snow properties in

the Eurasia (EU) and North America (NA) exhibit noticed discrepancy especially in snow density.
(Zhong et al., 2014;Bilello, 1984). One study pointed out that mean snow density in the former Soviet
Union (0.21 ~ 0.31 g/cm$^3$) was lower than the data from NA (0.24 ~ 0.31 g/cm$^3$) (Bilello, 1984), and
also Zhong et al. (2014) explained the possible reasons which resulting in the diversity of snow density
in EU and NA. Based on this, we separately constructed the SD retrieval models for EU and NA.

10       Step 6.2 Training dataset selection is the process of removing redundant features from spatial data.

The accuracy of estimated SD primarily depends on training data quality, which also demonstrate the
significance of the selection rule of training samples (Xiao et al., 2018). Inputting more data than
needed in the training dataset to train SD retrieval model, may lead to overfitting and an estimated SD
with high error. In this study, we collected an extremely large number of daily SD records over 25 years,
necessitating a optimized selection rule to avoid data information redundancy.

16       The selection rule proposed in previous research (Xiao et al., 2018) was modified and then it was

divided into two steps in here. Firstly, the numbers of sample in the three layers, layer1 (0≤SD<50),
layer2 (50≤SD<100) and layer3 (SD≥100), should be concretely quantified. To aviod an inflated
training sample in layer2 and layer3, we set a threshold (3 000) determined by several tests (not shown).
A threshold (12000) for layer1 was adopted following Xiao et al. (2018). Table 2 described the section
of training sample for each layer in detail. After that, the quality of training sample in each layers
determined by stratified random sampling is the second step. Stratification was performed in 1 cm SD
intervals. Note that, all the selecton operations in here were randomly performed.

24       Step 7. Through above steps, the daily estimated SD data in the Northern Hemisphere from

January 1992 to December 2016 (excluding July and August) were obtained. Owning to the nature of
radiometer observations, NHSnow products are only reliable in areas with seasonal dry snow cover.
Areas with sporadic wet or thin snow are not reliably detected and areas marked as snow-free may
include areas with wet snow. If one pixel is detected as snow cover by the detection decision tree
(Grody and Basist, 1996), but is likely to be shallow or medium-to-deep snow with an estimated value
of equal or less than 1 cm, the SD value is set as 5 cm (Che et al., 2016;Wang et al., 2008) (Fig. 4.).





Step 8. In this study, Greenland and Iceland are excluded from the generation and analysis of
NHSnow (NH_SD, NH_SWE) products due to their complex coastal topography and the difficulty in
discriminating snow from ice (Fig. 4) (Brown et al., 2010). Missing data and zero-data gaps occur in
the process of generating daily SD gridded products. Therefore, the following filters were applied.
Daily estimated SD was averaged with a sliding 7-day window to reduce noise and compensate for
missing data in the daily time series. For example, the SD estimate for 4 January is an average of the
assimilated scheme output for 1 to 7 January (Takala et al., 2011;Che et al., 2016). When finished, the
sliding SD method generated daily SD products for the entire Northern Hemisphere (NH_SD; Fig. 4).
**3.3 Estimation of SWE**
SWE contains more useful information for hydrologists than SD because it represents the amount
of liquid water in the snowpack available to the ecosystem as the snow melts. One way to estimate
SWE uses SD and snow density ($\rho$) as described in Eq. 2. Northern Hemisphere SWE products were
generated in this study using snow density that converts SD to SWE. (Eq. 2, Fig. 3 and 4, Step 9).

$$\text{SWE}(mm) = \text{SD}(cm) \times \rho(g/cm^3) \times 10 \qquad (2)$$

At present, the primary problem is to obtain relatively accurate snow density. In this study,
dynamical calculation methods were adopted to estimate snow density. Two methods are usually used
to convert SD to SWE. The first uses a fixed value, 0.24 g/cm$^3$ (or other value), without spatiotemporal
variation (Che et al., 2016;Takala et al., 2011). The second uses a temporally static by spatially variable
mask of snow density to estimate SWE and are used to generate current AMSR-E SWE products
(Tedesco and Narvekar, 2010). Since the snowpack are usually rather unstable, it is awfully
unreasonable to set the snow density in the whole snow season to a constant. Observations show that
snow density does evolve and tends to increase (decrease) throughout the snow season (from
September to June) (Dai et al., 2012;Sturm et al., 1995). Here, daily snow density is obtained following
Sturm et al.(2010) (Eq. 3).They used daily SD, day of the year (DOY), and the snow climate class (SC)
to produce snowpack bulk density estimates. In this method, knowledge of SC is used to capture field
environment variables (air temperature, initial density) that have a considerable effect on snow density
evolution.

$$\rho(\text{SD,DOY,SC}) = (\rho_{max} - \rho_0)[1 - exp(-k_1 \times SD - k_2 \times DOY)] + \rho_0 \qquad (3)$$

where $\rho_{max}$ is the maximum density, $\rho_0$ is the initial density, $k_1$ and $k_2$ are densification

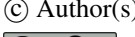



parameters for SD and DOY, respectively. $k_1$, $k_2$, $\rho_{max}$, $\rho_0$ vary with SC (Table 3). For operational
purposes in our work, DOY extend to 1 September each year (Matthew Sturm, personal
communication, 2018) running from −122 (1 September) to 181 (30 June). Sturm et al. (2010) didn't
compute snow density for the SC as ephemeral snow despite its presence in the Northern Hemisphere.
According to Zhong et al. (2014) study, the snow density of ephemeral is set to an fixed value, 0.25
g/cm$^3$. Finally, daily snow density is simulated by the Eq. 3 in the Northern Hemisphere during the
1992−2016 period.
**4 Results and Discussion**
**4.1 Snow depth**
**4.1.1 Validation of snow depth**
Here to give insight into relative performance of SD products, we compared three sources of snow
cover product (NHSnow, GlobSnow, and ERA-Interim/Land) with ground SD observations (Fig. 5-7)
using three indices bias, mean absolute error (MAE) and root mean square error (RMSE).The common
period (1992 - 2010) daily SD of three products (Section 2.4) were collected as validation data. This
validation work primarily focus on snow cover stabilization stage (December to February). Since the
snow density change slowly over a smaller range in snow cover stabilization stage (Xiao et al., 2018),
using a constant value (0.24 g/cm$^3$) for GlobSnow could introduce relative little error (Section 3.3).
Subject to the unavailability of SWE station observations, the evaluation of SWE can't be carried out.
The relatively little bias (blue and green dots) between the estimated SD from three products
against measured SD is located in mid and low latitude regions (< 60 °N) for these three snow depth
datasets (NHSnow, GlobSnow, and ERA-Interim/Land; Fig. 5). However, a large bias was found in the
polar region and along the coast, such as the north of Russia near the Arctic Ocean, Russian Far East,
Korean peninsula, Northern Mediterranean and Northeast Canada. For NHSnow and GlobSnow, most
bias is distributed near the μ=0 line with high frequency, although some bias is greater than 100 (or less
than −100) (Fig. 5b, d). Positive (negative) biases indicate mean grid cell values less (greater) than
those of the respective stations SD measures. Fig. 5c showed the ERA-Interim/Land overestimate snow
depth in Western Siberian Plains and Eastern European Plains (around 60 °N; orange dots). As





reference, Average SD pattern of three products in February (1992-2010) were also provided in
Appendix (Fig. A)
For analysis indexes, MAE and RMSE, the distribution of error points of NHSnow and GlobSnow
are much the same as the distribution of its bias (Fig. 5-7).We used all evaluation records to calculate
three precision indexes for three products. We found that the bias, MAE and RMSE is 0.59 cm, 15.12
cm and 20.11 cm, respectively, for NHSnow gridded product, but more bias (1.19 cm), MAE (15.98 cm)
and lower RMSE (15.48 cm) for GlobSnow (Table 4). This comparison (NHSnow vs. GlobSnow)
showed relatively good agreement, although NHSnow over- or underestimated the SD with larger
RMSE. Overall, the performance of GlobSnow was better than the NHSnow gridded product. However,
part of the validation data were also applied for GlobSnow assimilation, it is highly possible that in this
case GlobSnow validation may not completely independent. The different performance for these two
products may be mainly because the evolution of snow grain size by HUT (The Helsinki University of
Technology) model was used to generate SWE in GlobSnow. Che et al. (2016) reported that the grain
size is more important than snow density and temperature. Further, ERA-Interim/Land had the worst
performance of all three products with highest bias (-5.60), MAE (18.72) and RMSE (37.77). The
smallest bias is located near mid-latitude regions (< 50 °N) and much of the bias lies at 0–100 cm for
ERA-Interim/Land products (Fig. 5e, f). It must be noted that there are 89 bias records in two stations,
which located in Novosibirsk Islands and Victoria Island, is much less than -300 cm (approximately
-3000 cm). Large MAE and RMSE can be found in high latitude and coastal region (Fig. 5e). Unlike
NHSnow and GlobSnow, ERA-Interim/Land is more likely to overestimate SD and appears to be less
consistent with in situ observation across the Northern Hemisphere (Fig. 5f). Through analyzing ground
observation, we can see that deep snow is distributed in high latitude areas.
While the gridded products do a fairly good job of representing smaller accumulations of SD
(shadow and mid-deep snow cover), they all struggle to capture very high accumulations (deep snow)
with less bias, MAE and RMSE (Fig. 5-7, Fig. A). As a result, variation in snow cover could fail to be
adequately captured in areas with frequent deep snow and, thus, we should be cautious when
interpreting of this validation result.
Uncertainties in these three gridded snow products caused by ground temperature and topographic
factor could result in some level discrepancies between the measured and the estimated SD (Vander
Jagt et al., 2013;Snauffer et al., 2016). Forests exhibit strong influence on snow distributions by canopy



interception and the evolution of snow properties. The dense portions of boreal forests are widely
distributed in NA and northern EU (Friedl et al., 2010) Large bias, MAE and RMSE regions of three
gridded products (Fig. 5-7) cover vast areas of tall vegetation (forests and shrub). Furthermore, the
spatial inhomogeneity cause one grid cells (~25 km) that is almost not possible to completely cover by
one vegetation type (low heterogeneity). Because the estimated SD of NHSnow depends on land cover
types, this discrepancy induced by surface cover heterogeneity could partly account for why NHSnow
has a smaller MAE and RMSE for low vegetation (bare-land and prairie) distributed at middle and low
latitudes, than the higher vegetation (shrub and forest) areas at higher latitudes (Xiao et al., 2018).
As well, there are scale mismatches between in situ observation and the gridded products with
regard to snowpack properties and their spatiotemporal representativeness (Frei et al., 2012). It is
difficult to precisely validate coarse-resolution satellite observation using ground truth. Subsequently,
over- or underestimates are inevitable when using a single in situ (SD or SWE) observation to test the
veracity of the gridded products (Mudryk et al., 2015;Xiao et al., 2018). Snow surveys would benefit
from multiple measurements at different points within one pixel (López-Moreno et al., 2011). In situ
observations are highly representative when the SD varies smoothly in space, and poorly representative
when the SD is spatially stepped (Che et al., 2016). However, there is almost always a lack of sufficient
ground-measured data. To date, field site observations are still to be more authentic and reliable
datasets than satellite observation.
As a whole, the accuracy of estimated SD in the Northern Hemisphere presented a spatial
heterogeneity. Issues of scale and spatial heterogeneity of validation data notwithstanding, these
comparisons conducted in our work can yield valuable insight into the performance of these products.
**4.1.2 Variation of snow depth**
To better understand and interpret snow cover variation across the Northern Hemisphere, we
conducted an analysis of SD variation using seasonal maximum SD from 1992–2016. According to the
rules of variation level grading, which was divided into 5 grade (extremely significant increase,
significant increase, non-significant change, extremely significant decrease, and significant decrease;
Table 5), we can easily gained seasonal maximum SD variation level range 1992 to 2016. Figure 8
shows the variation pattern of seasonal maximum SD in three seasons (fall, winter and spring) with





statistical significance level. In three seasons, variation trend of seasonal maximum SD exhibited a
distinctly different pattern over the Northern Hemisphere since 1992. Seasonal maximum SD variation
results in fall illustrated that a reduction trend account for most area of the EU with the rate ranging
from 0 to 1 cm yr.$^{-1}$. The Figure 8a show the significant level pattern of corresponding maximum SD
change trend. We can find that the area, which show extremely significant decrease in fall, are mainly
located in the Russian Far East, the Qinghai-Tibet Plateau, the southern Siberian Plateau, and the
northeastern region of Canada. On the contrary, Russia's Taimer Peninsula and the United States'
Alaska region shows extremely significant increase trend (0 ~1 cm yr.$^{-1}$). In addition, the maximum SD
in winter and spring also exhibited extremely significant decrease in the Qinghai-Tibet Plateau and the
northeastern region of Canada as shown in Figure 8b and 8c. The area with extremely significant
decrease trend extent add a Western Siberian plain region. Wang and Li (2012) used nearly 50a of daily
station SD observation data to analyze the trend of maximum SD in China. The variation trend of
seasonal maximum SD in the Qinghai-Tibet Plateau form previous study is consistent with the
conclusion observed in this study (Wang and Li, 2012).There are more regions in seasonal maximum
SD with extremely significant increase trend in winter and spring (green region). Furthermore, a
strange phenomenon that the variation trend of seasonal maximum SD in the Russian Far East show
extremely significant decrease, while it is in inverse in spring. This variation trend of maximum SD in
spring analyzed using NHSnow products is consistent with the analysis results using GlobSnow
products from recently published study (Wu et al., 2018). It need be pointed out that the significant
increase (decrease) area is located around extremely significant increase (decrease) as shown in Figure
8. No matter which season, although the variation trend of maximum seasonal SD didn't pass the
significance level test, we can draw the conclusion that the wide range of area across the Northern
Hemisphere experienced pronounced change during the period 1992 to 2016.
Finally, we analyzed season variation analysis of SD across the Northern Hemisphere using
seasonal average SD as analysis index. Seasonal average SD was defined as the cumulative SD divided
by the days in one snow cover season.SD variation rate fluctuated in different regions and seasons. It
was generally large in the region north of 55° N (Fig. 9, Fig. B and C in appendix). This fluctuation
was large in winter with high of $-0.11 \pm 0.40$ cm yr.$^{-1}$ than other seasons during 1992–2016 (Fig. 9d,
Table 6.), which means that the maximum changes occurred in winter. Similar conclusion also can be
easily found in the two periods 1992–2001 and 2002–2016 (Fig. B-d, C-d and Table 6). Although not





all variation trends passed the significance test, most regions in the Northern Hemisphere show

increasing trends during 1992-2001 (Fig. B; Table 6). The SD variation trend in the three seasons

during 2002–2016 was reversed. The SD absolute variation rate during 2002–2016 is apparently greater

than its rate during 1992–2001 (Fig. C; Table 6). The last century were considered to be the warmest

period.

The high fluctuation of SD variation rate especially occurred in the polar region (the arctic and the

Tibetan plateau) for three seasons. In the context of global climate change, we found that winter SD

variation was more sensitive to climate change (Brown et al., 2010). The strength of this relationship is

spatially complex, varying by latitude, region, and climate condition.

**4.2 Total SWE**

GlobSnow dataset covers all land surface areas excluding mountainous regions, glaciers and

Greenland as described in Section 2.4. From above analysis, we can find that ERA-Interim/Land have

somewhat poor performance in SD estimation. Thus, further analysis of snow cover variation in the

Northern Hemisphere used NHSnow products as analysis data. The forecast for total SWE (or snow

mass) have great potential consequences on agriculture practices in many regions. Total SWE in here is

calculated by SWE multiplied by snow cover area (Qin et al., 2006). It should be noted that the snow

classification tree (Grody and Basist, 1996), which have been applied in many studies (Che et al.,

2008;Dai et al., 2017;Yu et al., 2012), was used to detect snow cover for NHSnow product. Liu et al.

(2018) also reported that Grody's algorithm had higher positive predictive values and lower omission

errors by testing snow cover mapping algorithms with the in situ SD over China. In this study, Annual

(or monthly average) total SWE, which is the sum of daily (or the mean of monthly) total SWE in one

snow cover year (or each month of 25 years).

Interannual variation (Fig. 10) and intra-annual cycles (not show figure) of total SWE over the

Northern Hemisphere were used to analyze total SWE variation characteristic over the past 25 years

(1992–2016). Figure 8 depicts the time series of interannual variation of total SWE anomaly with

respect to 1992–2016 reference period. The maximum anomaly occurred in 1998–1999 period with

the minimum was during 2015–2016. It present particularly significant decreasing trends (P ≤ 0.05)

during 1992–2016, at the rate of approximately -5794 km$^3$ yr.$^{-1}$. Trend analysis reveals that total SWE

have 12.5% reduction from 1992 to 2016. There is a slow variation rate by about 710 km$^3$ yr.$^{-1}$ (P >

0.05) for 1992-2001 period. In contrast, the total SWE anomaly significant decrease (P ≤ 0.05) after
2002 at rate of approximately -9041 km$^3$ yr.$^{-1}$, which may lead to a decreasing trends of total SWE
during 1992–2016. There was a sudden drop of total SWE in 2008–2009 as found in previous studies
(Derksen and Brown, 2012;Wang et al., 2018). However, other factors, for instance, oceanic and
atmospheric heat transport, sea ice season wind, and solar insolation anomalies, may have contributed
to the fluctuation of total SWE (Liu and Key, 2014). Variation of total SWE across the Northern
Hemisphere could well capture the variation characteristic of the Arctic sea ice extent (Tilling et al.,

8   2015).

9       When analyzing long-term variation of monthly average total SWE, ten months (September to

June) exhibit significant decreasing apart from March and April (Table 7). The maximum decrease
was approximately -1066 km$^3$ yr.$^{-1}$ in January while the minimum decrease occurred in September at
-177 km$^3$ yr.$^{-1}$. An increasing trend appears in March with a rate of approximately 68 km$^3$ yr.$^{-1}$ (P >
0.05), however, relatively large decrement in fall and winter are unable to partially be offset by the
increment of March. Compared with the fall (September to November) and spring (February to June),
the interannual variability of monthly average total SWE significantly decreased in winter (December
to January), with average rate of less than -1000 km$^3$ yr.$^{-1}$. We also found that the monthly average
total SWE reduction fluctuated ranging from -66% to -4% for each month (September to June) over
1992-2016 (Table 7). The largest and smallest reduction were about 65.8% and 4.2%, which occurred
in June and March, respectively.
Over large areas, it is extremely convenient to use remote sensing to infer SWE. Albeit there are
numerous ways to estimate SWE, it is very challenging to determine precise distributions of SWE at
regional and global scales (Chang et al., 1987;Kongoli, 2004;Tedesco and Narvekar, 2010;Bair et al.,
2018). Snow density, which can be used to convert SWE from SD, is potential and key factor in
accurate estimation of SWE (Sturm et al., 2010;Tedesco and Narvekar, 2010). In fact, snow density
typically varies from 0.05 g/cm$^3$ for new snow at low air temperatures to over 0.55 g/cm$^3$ for a ripened
snowpack (Anderton et al., 2004;Cordisco et al., 2006). Noteworthily, this study using dynamic snow
density to convert SD to SWE is based on the assumption that snowpack occurs as a single layer
(Sturm et al., 2010), to capture dynamic characteristics of snow property. The evolution of the
ephemeral snow class did not be provided by Sturm et al. (2010). The mean value (0.25 g/cm$^3$) of snow
density of ephemeral snow (Zhong et al., 2014), which mean that without any evolution throughout the



snow cover year. Meanwhile, this value for ephemeral snow was set as 0.2275 $g/cm^3$ in Tedesco and
Jeyaratnam (2016) study. Snow density also exhibits great heterogeneity in vertical direction, so that a
single layer of snow concept cannot fully capture the snowpack property. The density of the top
snowpack (fresh snow; ~ 0.10 $g/cm^3$) increases gradually from the top toward the bottom (Dai et al.,
2012). The bottom layer of snowpack is old undergoing compaction and grain size growth with a
relatively high density (0.3~0.6 $g/cm^3$). Although our snow density description strategy does not
completely describe the actual evolution in snow density, there is no better alternative.
**4.3 Snow cover days**

9        Snow cover days (SCD) is defined as the number of days in one snow cover year in which SD is

over 0 cm (Zhong, 2014). Snow cover year was defined as the period between July of a given year and
June of the following year (Xiao et al., 2018). A least-squares regression was used to analyze the
variation of SCD for each pixel from 24 snow cover years, with per-pixel evaluation of significance
(F-test).

14       We exploring the variation in SCD during 1992-2016. Most areas across the Northern Hemisphere

present a prominently decreasing trend at a rate ranging from 0 to 5 day $yr.^{-1}$ (Fig. 11a). Decreasing
regions are mainly distributed in EU. For example, north of Russia and large parts of central Asia. The
area that shows decreasing trends of SCD in EU is much larger than that in NA (Fig. 11a) (Derksen and
Brown, 2012). Areas that the decrease at a rate greater than 5 day $yr.^{-1}$ are almost all located in China,
such as North of Qilian Mountain, central Tibetan Plateau, and Tianshan Mountain. Areas that exhibits
increasing trends, can be found in central of NA, Western Europe, Northwestern Mongolia, and some
parts of China. Throughout the Northern Hemisphere (Fig. 11b), the decreasing trend covered most
parts of the regions (25 ~ 85 °N) with a mean decreasing rate of approximately 1.0 day $yr.^{-1}$. Latitudes
around 50 °N is an exception where variation is close to 0 day $yr.^{-1}$. The most notable variation trend
(decreasing or increasing) occurred over polar region (Fig. 11b). This may be because there are few
pixels in the polar mainland.

26       SCD variation rate also were divided into 5 grade (Table 5). Unlike SCD variation rate patterns,

the variation level pattern shows that the non-significant changes area dominates SCD variation trends
across the Northern Hemisphere (Fig. 11c). Extremely significant and significant decrease appear in
northwest of Hudson Bay in Canada, Kamchatka peninsula, Eastern European plains, the north of



Russia, Iranian plateau, and several regions in China (the Tibet Plateau, Tianshan Mountain and
Northeast China Plain). In addition, extremely significant and significant increase only occur in a
limited area of NA, eastern Tibet Plateau regions, and China's central and northern regions.

4         Interestingly, the opposite variation trends in SCD and SD appear in several regions. Maximum

SD in spring (Fig. 8c) and annual average SD (figure not shown) show extremely significant increasing
trends , whereas SCD exhibit extremely significant decreases in corresponding regions (Fig. 11c), such
as Central Siberian Plateau, Greater Khingan Mountains in China, and the eastern Scandinavian
Peninsula. This different variation trend of SD and SCD was also reported by Zhong et al. (2018) using
ground-based data. The primary reason may be the increase of frequency of extreme snowfall in which
SD could demonstrate on increasing trend. Additionally, a recent study found that the greater SWE, the
faster melting rate leading to a shortened SCD in Northern Hemisphere (Wu et al., 2018).

12        Despite the similarities between the station- and satellite-derived time series, it can be

demonstrated that Northern Hemisphere meteorological station data do not provide perfect large-scale
variation characteristics of ground snow cover (Zhong et al., 2018). Our analyses provide further
evidence supporting observations of significant decreasing trends in SCD occurring in the Northern
Hemisphere. Compared to SCD derived from optic sensors snow cover product, however, the specific
quantity of SCD and SCD variation rate derived from NHSnow SD data was overestimated (Wang et
al., 2018;Hori et al., 2017). The SCD variation trends derived from NHSnow product almost is same as
derived from optical snow cover product in variation pattern (Hori et al., 2017).

20        Since the optical (MODIS or AVHRR) and microwave sensors (SSM/I or AMSR-E) respond in

different parts of the electromagnetic spectrum, the estimated snow cover will to be somewhat vary.
The shallow snow could not induce volume scattering at 37 GHz, and thus passive microwave
observations often give better snow cover result at thick snow (>5 cm) (Foster et al., 2009;Wang et al.,
2008). The threshold for SCD definition in here is 0 cm, whereas it is 1 cm or larger in other studies
(Ke et al., 2016;Dyer and Mote, 2006). As well, another explanation for these discrepancy could be
snow cover identification algorithm (Liu et al., 2018;Hall et al., 2002).

27        The microwave radiation characteristics of snow cover is similar to that of precipitation, cold

desert and, frozen ground (Grody and Basist, 1996). Commission and omission errors in NHSnow
product may result from coarse spatial resolution, snow characteristics and topography according to
Dai et al. (2017), precipitation (Liu et al., 2018;Grody and Basist, 1996) especially over frozen ground



(Tsutsui and Koike, 2012). Algorithm several rules were used to distinguish snow from precipitation,
cold desert, and frozen ground (Xiao et al., 2018), it is impossible to entirely remove interference
factors in each image. Additionally, the precondition of NHSnow is dry snow, which mean almost no
wet snow was considered into SCD variation analysis (Singh and Gan, 2000). The poorer performance
of the microwave derived products was anticipated because of documented difficulties monitoring
snow cover over forested and mountainous terrain (Vander Jagt et al., 2013;Smith and Bookhagen,

7    2016).

**5 Conclusions**

9       This project applied the SVR SD retrieval algorithm proposed by Xiao et al (2018), which using

PM remote sensing and other auxiliary data, to develop a long term (from January 1992 to December
2016) Northern Hemisphere daily SD and SWE products (NHSnow) with 25-km spatial resolution. We
then analyzed the spatial and temporal change in snow cover (SD, total SWE and, SCD) across the
Northern Hemisphere, and quantified the magnitude of variation of snow cover using SD and SWE
extracted from NHSnow product.

15       In this study, we validated and compared among daily gridded products (NHSnow, GlobSnow and

ERA-Interim/Land) against ground snow-depth observations. The results show relatively high
estimation accuracy of SD from NHSnow, providing the relatively little bias, RMSE, and MAE
between the newly SD products and in situ observation. Analysis of SD variation revealed that the
variation rate ranging from 0 to 1 cm yr.$^{-1}$ (negative and positive) dominates the change in the Northern
Hemisphere, and the maximum changes appear in winter. Additionally, the results revealed the overall
SD trends in three seasons show increasing trend during 1992–2001, however it has a decreasing trend
during 2002–2016. Similar conclusions also appear in total SWE change analysis. The total SWE
shows a 12.5% reduction and the monthly average total SWE is 65.8% for the largest reduction and a
4.2% for least reduction which occur in June and March, respectively. The total SWE report
well-documented significant decreasing trends ($P < 0.05$) during the study period. Regression analysis
multi-year Northern Hemisphere SCD exhibits a prominent decreasing trend at a rate ranging from 0 to
5 day yr.$^{-1}$. The area of decreasing trends of SCD in EU is much larger than in NA. Unlike the SCD
variation rate, its variation level shows that non-significant changes areas dominate the variation



pattern across the Northern Hemisphere. An abnormal and interesting phenomenon is that opposite
SCD and SD variation trends appear in several regions.
While this study shed light on the spatiotemporal variability trends of snow cover across the
Northern Hemisphere using 25-year NHSnow product, we cannot claim NHSnow dataset could
completely capture the climate change signal in each region and season. Because of the deficiencies
and limitations (e.g. overestimation, underestimation), further efforts should be made to improve the
estimation accuracy and robustness of the SD inversion algorithm. Additionally, when more reliable
and numerous data become available, we will do more comprehensive validation over higher latitudes
and mountainous regions (Dai et al., 2017). Meanwhile, the validation analysis also should be carried
out in complex terrain and different land cover types (Tennant et al., 2017;Snauffer et al., 2016). It is
recommended that future work focus on the climatic effects and climatological causes in snow cover
changes to comprehensively understand the associated snow cover change mechanisms against a
climate change background (Huang et al., 2017;Flanner et al., 2011;Cohen et al., 2012).
**Acknowledgments**
This study was funded by the National Natural Science Foundation of China (grant nos. 91325202;
41871050; 41801028), National Key Scientific Research Program of China (grant no. 2013CBA01802),
and the Strategic Priority Research Program of Chinese Academy of Sciences (grant nos.
XDA20100103; XDA20100313).





1    **Appendix**

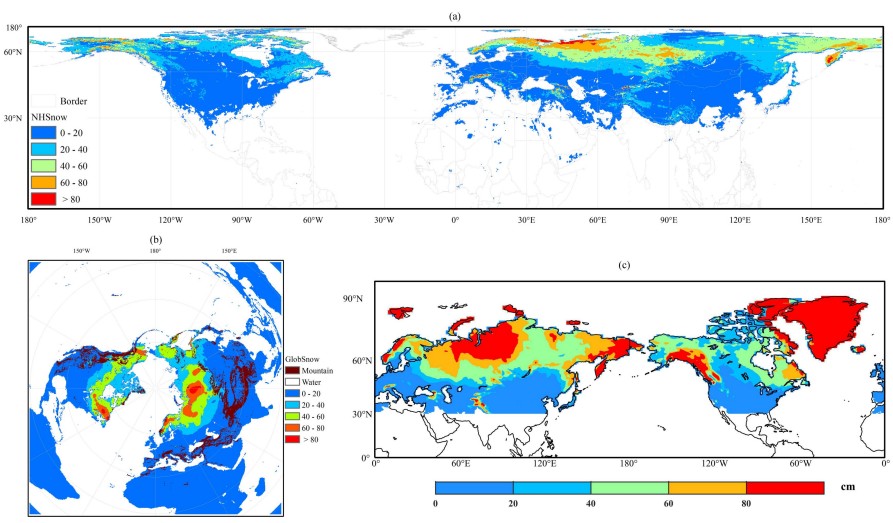

3    Figure A. Monthly average snow depth climatology of three products in February during 1992-2010: a)

4                                      NHSnow; b) GlobSnow, c) ERA-Interim/Land

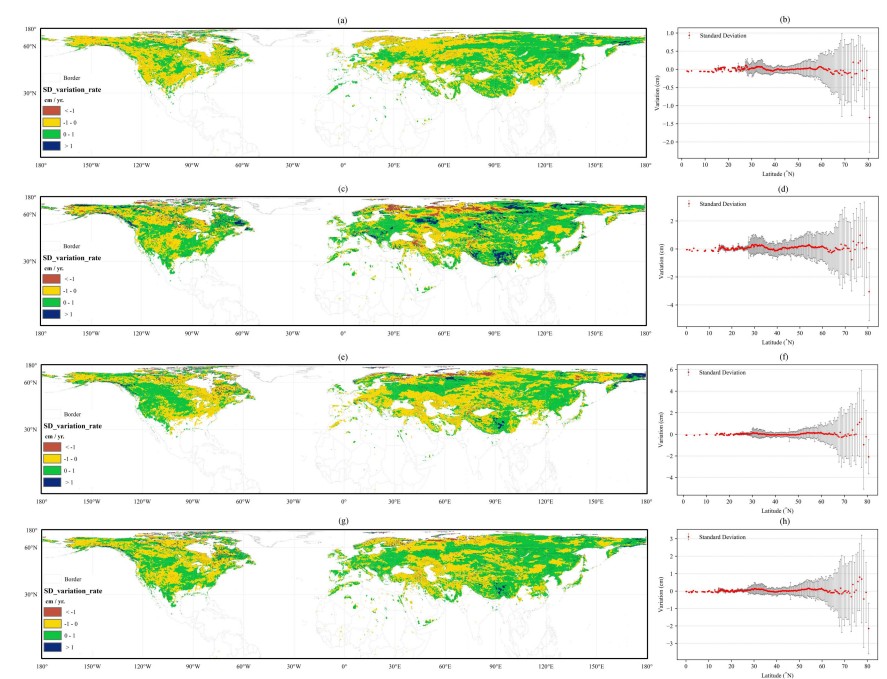





Figure B. The variation rate pattern of annual average (season) SD over the Northern Hemisphere for
three snow cover season, fall (a, b; September to November), winter (c, d; December to February),
spring (e, f; March to June) from 1992-2001. Black dots in (a, c, e, g) indicate that the changes are
significant at 95% confidence level (CL). The zonal distribution in (b, d, f, h) are mapped at 0.25
degree resolution in latitude. The error bars in (b, d, f, h) is one times of standard deviation.

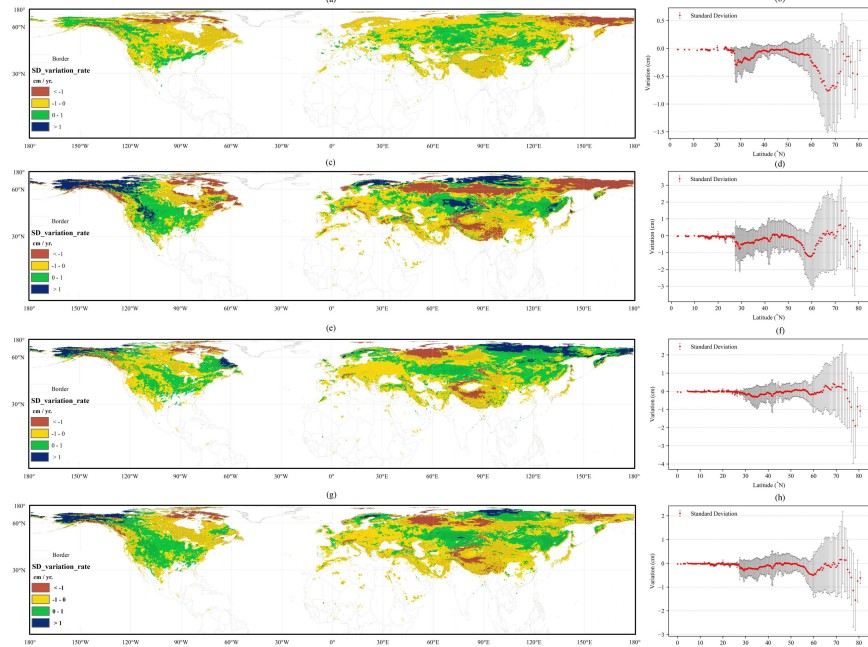

Figure C. The variation rate pattern of annual (season) average SD over the Northern Hemisphere for
three snow cover season, fall (a, b; September to November), winter (c, d; December to February),
spring (e, f; March to June) from 2002-2016. Black dots in (a, c, e, g) indicate that the changes are
significant at 95% confidence level (CL). The zonal distribution in (b, d, f, h) are mapped at 0.25
degree resolution in latitude. The error bars in (b, d, f, h) is one times of standard deviation.



1         Table A. AVHRR Global Land Cover classification and reclassification schemes

| Value | Classification Label | Reclassification Label |
|-------|----------------------|------------------------|
| 0 | Water | Water |
| 1 | Evergreen needle leaf forest | Forest |
| 2 | Evergreen broad leaf forest | |
| 3 | Deciduous needle leaf forest | |
| 4 | Deciduous broad leaf forest | |
| 5 | Mixed forest | |
| 6 | Woodland | |
| 7 | Wooded grassland | Prairie (Grassland) |
| 10 | Grassland | |
| 8 | Closed shrub land | Shrub |
| 9 | Open shrub land | |
| 11 | Cropland | Bare-land |
| 12 | Bare ground | |
| 13 | Urban and built | |



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





**List of Tables and Figures**
Table 1 Detail description for SSM/ and SSMIS sensors. H and V denotes horizontal and vertical
polarization, respectively.

| Satellite | SSM/I | | SSMIS |
|---|---|---|---|
| Platform | F 11 | F 13 | F 17 |
| Temporal coverage | 1991.12-1995.5 | 1995.5-2008.6 | 2006.12 - |
| Channels (GHz) | 19 H, V; 22 V; 37 H, V; 85 H, V | | 19 H, V; 22 V; 37 H, V; 91 H, V |

Table 2. Training sample filter rules

| Layer ID | Filter rules |
|---|---|
| Layer2. | If $\text{Number}_{total}(layer2) \leq 3000$ <br><br> $\text{Number}_{training}(layer2) = (\text{Number}_{total}(layer2))/2$ <br><br> Else $\text{Number}_{training}(layer2) = 3000$ |
| Layer3. | If $\text{Number}_{total}(layer3) \leq 3000$ <br><br> $\text{Number}_{training}(layer3) = (\text{Number}_{total}(layer3))/2$ <br><br> Else $\text{Number}_{training}(layer3) = 3000$ |
| Layer1. | If $\text{Number}_{training}(layer2) > 2000$ or $\text{Number}_{training}(layer3) > 1000$ <br><br> $\text{Number}_{training}(layer1)$ <br><br> $= 15000 - \text{Number}_{training}(layer2) - \text{Number}_{training}(layer3)$ <br><br> Else $\text{Number}_{training}(layer1) = 12000$ |

Table 3 Snow density estimation model parameters

| Snow class | $\rho_{max}$ | $\rho_0$ | $k_1$ | $k_2$ | References |
|---|---|---|---|---|---|
| Alpine | 0.5975 | 0.2237 | 0.0012 | 0.0038 | |
| Maritime | 0.5979 | 0.2578 | 0.0010 | 0.0038 | |
| Prairie | 0.5940 | 0.2332 | 0.0016 | 0.0031 | Sturm et al. (2010) |
| Tundra | 0.3630 | 0.2425 | 0.0029 | 0.0049 | |
| Taiga | 0.2170 | 0.2170 | 0 | 0 | |
| Ephemeral | 0.2500 | 0.2500 | 0 | 0 | Zhong et al. (2014) |



1    Table 4. The evaluated indexes (bias, MAE, RMSE; unit: cm) for three gridded SD products (NHSnow,

2                              GlobSnow, ERA-Interim/Land).

| Products | Bias | MAE | RMSE |
|---|---|---|---|
| NHSnow | 0.59 | 15.12 | 20.11 |
| GlobSnow | 1.19 | 15.98 | 15.48 |
| ERA-Interim/Land | -5.60 | 18.72 | 37.77 |

4                              Table 5. Rules of variation level grading

| Variation rate | P value | Variation level |
|---|---|---|
| rate > 0 | $p \leq 0.01$ | extremely significant increase |
| rate > 0 | $0.01 < p \leq 0.05$ | significant increase |
| - | $P > 0.05$ | non-significant change |
| rate < 0 | $p \leq 0.01$ | extremely significant decrease |
| rate < 0 | $0.01 < p \leq 0.05$ | significant decrease |

6    Table 6. Mean variation rate of average SD (cm yr.[-1]) over the Northern Hemisphere for three common

7    period (1992-2016, 1992-2001, 2002-1996) and snow cover seasons (fall, winter, spring). Std. means

8                              standard deviation

| Season | 1992-2016 (Mean ± 1 Std.) | 1992-2001 (Mean ± 1 Std.) | 2002-2016 (Mean ± 1 Std.) |
|---|---|---|---|
| Fall | -0.08 ± 0.11 | -0.01 ± 0.19 | -0.15 ± 0.22 |
| Winter | -0.11 ± 0.40 | 0.06 ± 0.62 | -0.22 ± 0.75 |
| Spring | -0.04 ± 0.25 | 0.02 ± 0.51 | -0.07 ± 0.41 |
| Year | -0.06 ± 0.20 | 0.02 ± 0.35 | -0.11 ± 0.34 |

10    Table 7. Variation rate and changes of monthly average total SWE. The asterisk indicate that the

11                              changes are significant at 95% confidence level

| Month | Variation rate (km$^3$/yr.) | % Change in the mean of monthly average total SWE over 1992-2016 period |
|---|---|---|
| September | -176.66* | -63.73% |





| October | -776.92* | -43.95% |
| November | -1060.10* | -26.83% |
| December | -979.71* | -4.82% |
| January | -1065.72* | -9.53% |
| February | -838.79* | -9.52% |
| March | 67.54 | -4.17% |
| April | -128.04 | -6.44% |
| May | -343.55* | -20.34% |
| June | -226.01* | -65.79% |

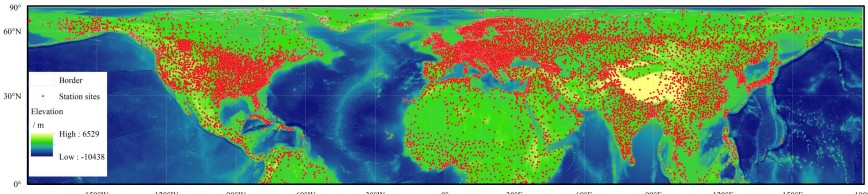

4    Figure 1. Distribution of Meteorological stations overlaid on ETOPO1 in the Northern Hemisphere.

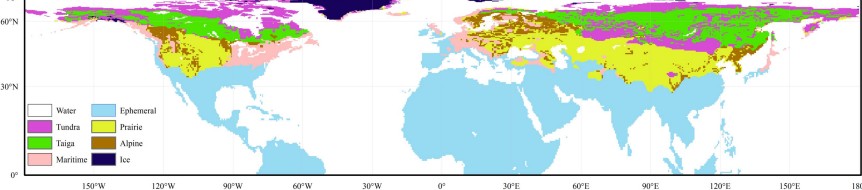

7    Figure 2. Snow Class distribution in the Northern Hemisphere



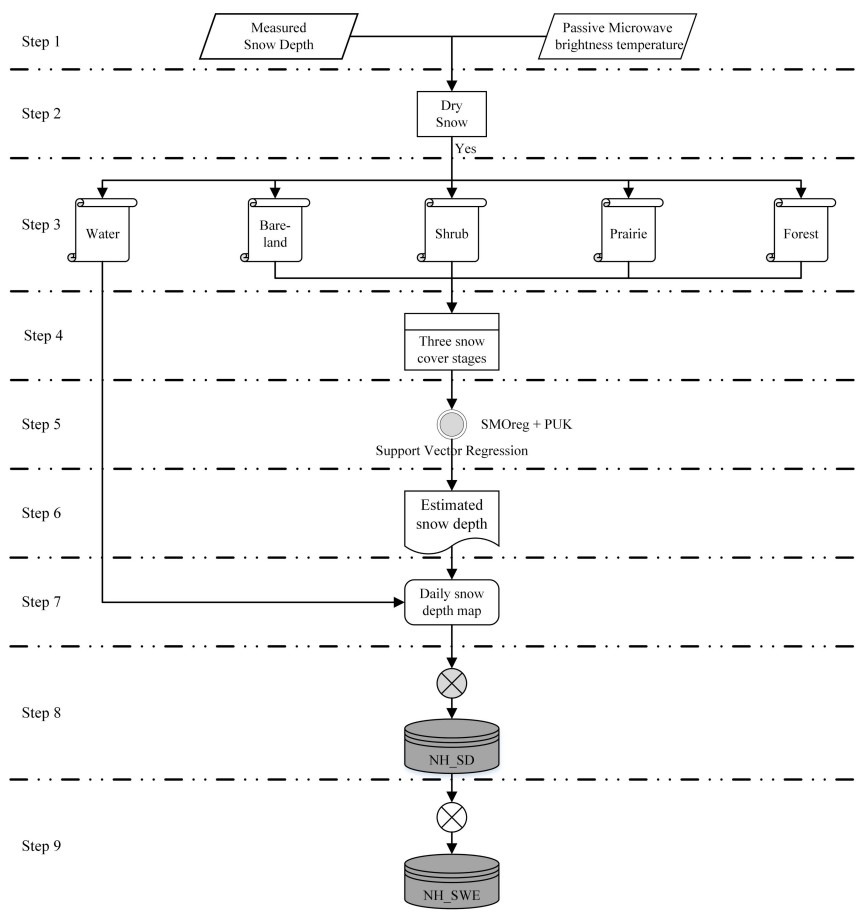

2      Figure 3. Process flowchart diagram for developing Northern Hemisphere daily snow depth and snow

3                  water equivalent data



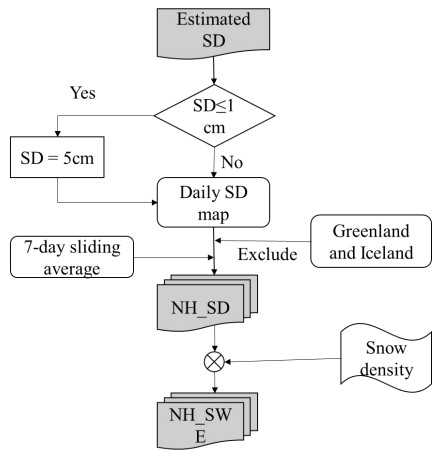

2          Figure 4. Flowchart diagram of the generation of NHSnow products.

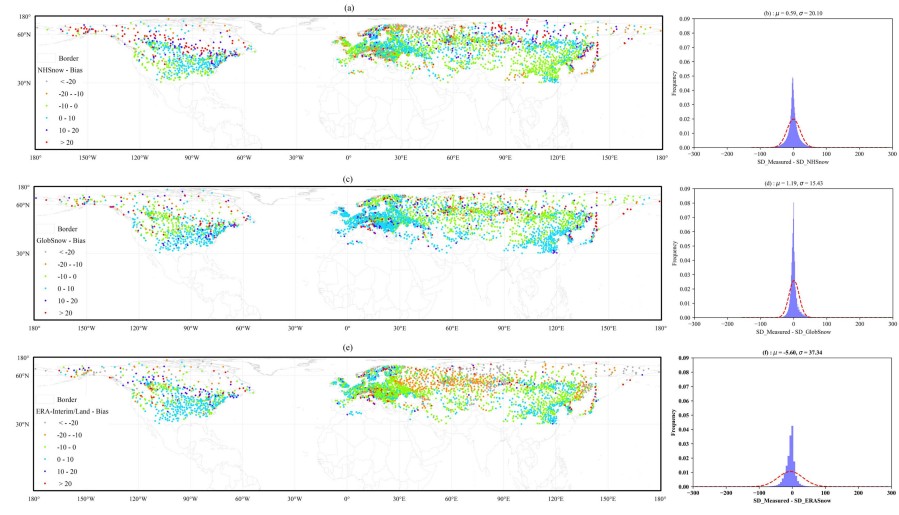

4          Figure 5. Bias of each meteorological station and histogram of biases for three products: a), b)

5     NHSnow; c), d) GlobSnow, e), f) ERA-Interim/Land. The red dashed line in right column figures are

6                          the fitted normal distribution curve



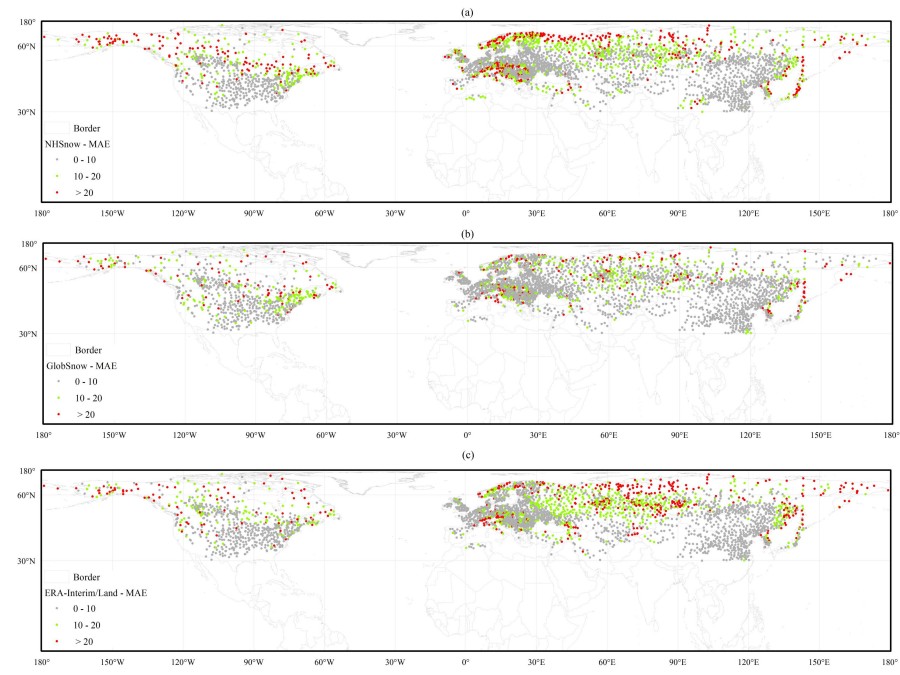

2    Figure 6. MAE of each meteorological station for three products: a) NHSnow, b) GlobSnow, c)

3                        ERA-Interim/Land.

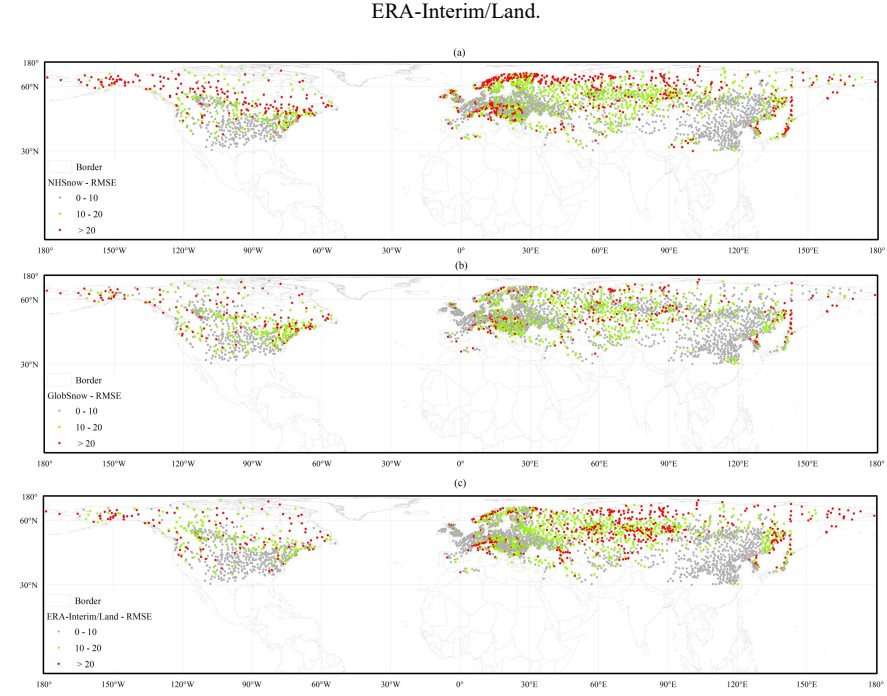

5    Figure 7. RMSE of each meteorological station for three products: a) NHSnow, b) GlobSnow, c)



1                                 ERA-Interim/Land.

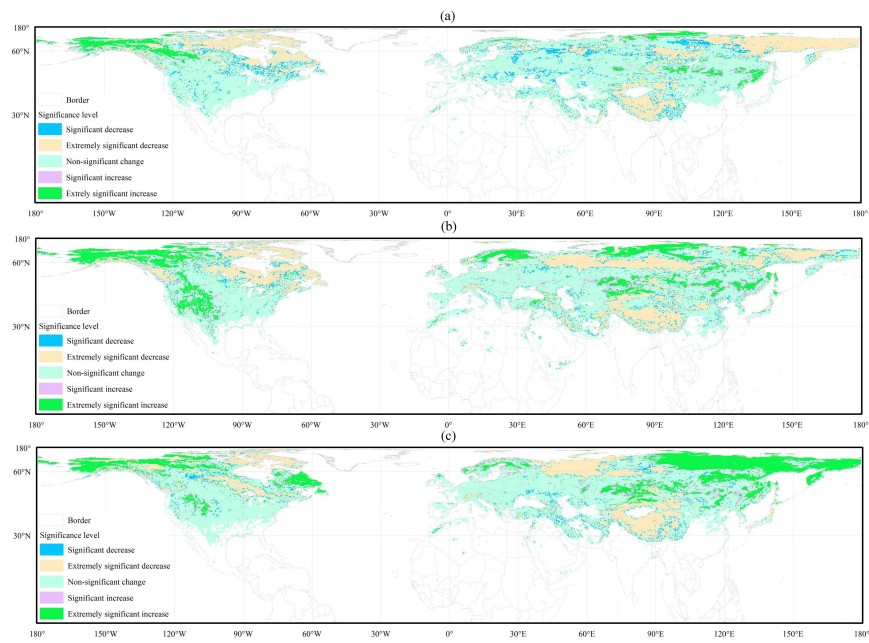

Figure 8. The variation rate pattern of season maximum SD with statistical significances over the
Northern Hemisphere for three snow cover season, fall (a; September to November), winter (b;
December to February), spring (c; March to June) from 1992-2016.

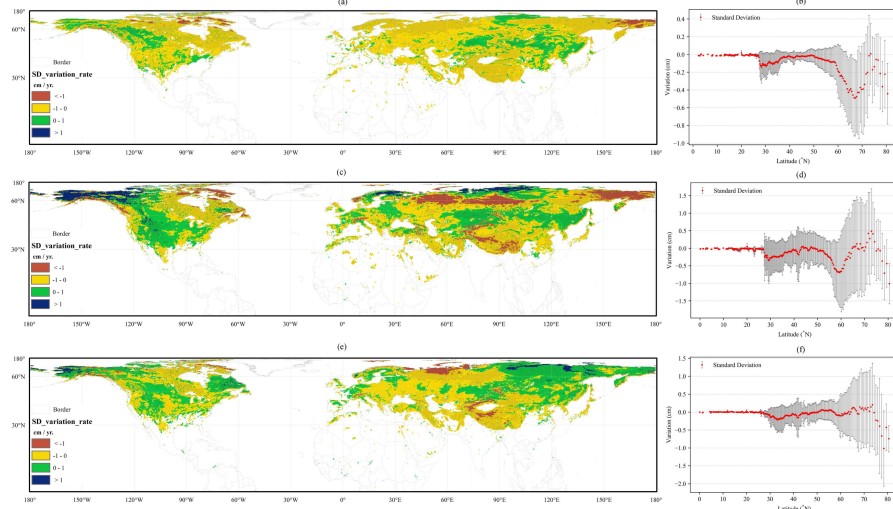

Figure 9. The variation rate pattern of season average SD over the Northern Hemisphere for three snow




cover season, fall (a, b; September to November), winter (c, d; December to February), spring (e, f;
March to June) from 1992-2016. Black dots in (a, c, e) indicate that the changes are significant at 95%
confidence level (CL). The zonal distribution in (b, d, f) are mapped at 0.25 degree resolution in
latitude. The error bars in (b, d, f) is one times of standard deviation.

$$y_a = -5793.65 * x_a + 1.16 \times 10^7$$
$$p <= 0.05, R^2 = 0.75$$

$$y_c = -9041.48 * x_c + 1.81 \times 10^7$$
$$p <= 0.05, R^2 = 0.82$$

$$y_b = 710.089 * x_b - 1.38 \times 10^6$$
$$p > 0.05, R^2 = 0.01$$

Figure 10. Interannual variation of total SWE over the Northern Hemisphere for three period
1992-2016 (black line), 1992-2001 (blue line), and 2002-2016 (red line), with respect to the 1992-2016
mean value. Trends estimates were computed from least squares. P is the confidence level for the
coefficient estimates, $R^2$ is the goodness of fit coefficient.

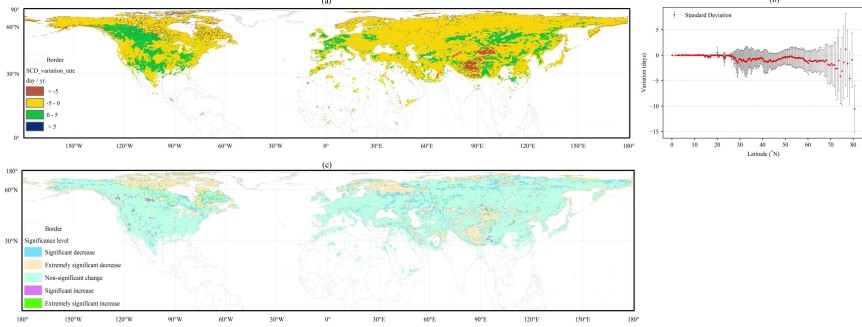

Figure 11. The variation rate pattern of SCD (a) and their statistical significances (c) over the Northern



1    Hemisphere from 1992-2016. The zonal distribution in (b) are mapped at 0.25 degree resolution in

2         latitude. The error bars in (b) is one times of standard deviation.

