# Peer review of "Spatiotemporal variation of snow depth in the Northern"

_The Cryosphere, 2019_

## Referee Comment (RC1) · Anonymous Referee #1 · 3 May 2019

In this manuscript, the authors use a support vector regression (SVR) algorithm that they developed in a previous paper to estimate snow depth from passive microwave observations. In addition to evaluating their estimates of snow depth against values from GlobSnow and ERA-Interim/Land, they also use snow density assumptions to estimate snow water equivalent (SWE) for the Northern Hemisphere. Their major conclusion is that SWE has been declining by ~5 800 km3 a year, or approximately 139 200 km3 over their 24-year study period. The authors say this decline is equivalent to a 12.5% reduction of SWE over the study period, suggesting the initial amount of SWE was ~1 113 600 km3.

I believe there is a fundamental flaw in how the authors are calculating annual snow accumulation in this manuscript. Their estimate of annual SWE is orders of magnitude larger than other global datasets suggest. Mudryk et al. (2015) show that the Northern Hemisphere has an average annual snow accumulation of 3500 km3 (see Figure 1a, taken from Figure 3 in that manuscript). Using four commonly used global datasets (ERA-Interim, GLDAS, MERRA2, and VIC), I estimate the long-term-average global snow storage to be ~4000 km3 (see Figure 1b). Even if these global models/reanalyses are underestimating SWE, it is unlikely they are wrong by as much as this manuscript indicates. I believe the authors may be summing daily values of SWE when calculating their annual total SWE, as one would do when calculating annual precipitation from daily precipitation values. However, this is incorrect when working with SWE. Instead, the authors should consider comparing the annual maximum SWE over their period of record. This will not lead to such a dramatic value of SWE decline, but I think it would be interesting to see how their method compares to changes in SWE from GlobSnow, ERA-Interim/Land, and other global data products.

With this mistake, the manuscript is not ready for publication. But if the authors redo their SWE calculations and the following analyses, I would be interested to see the SWE results from their SVR method. Since this error is critical to the main conclusions of the manuscript, I do not include a review of the rest of the paper.

Reference: Mudryk, L. R., Derksen, C., Kushner, P. J., and Brown, R.: Characterization of Northern Hemisphere Snow Water Equivalent Datasets, 1981–2010, Journal of Climate, 28, 8037-8051.

[Figure]

[Figure]

Figure 1. (a) Northern Hemisphere snow mass, in kg, from Mudryk et al. (2015). (b) Global snow water storage, in km3, from four global data products.

**Fig. 1.**

---

## Author Comment (AC1) · 9 May 2019

- 1 Dear editor and reviewer,
- 2 Thank you for your positive comments and very important recommendations to improve our
- 3 manuscript. We have carefully modified the manuscript based on your suggestions and provide a
- 4 response to each comment. Reviewer comments are given in black, and responses are given in
- 5 blue. Below we provide a marked-up manuscript version showing the changes based on your
- 6 comments. The main modifications to the manuscript are as follows:
- 7 1. Fig. 10 and Table 7 were revised according to your suggestions.
- 8 2. We revised the description in Abstract, Section 4.2 and, Section 5 accordingly.
- 9 3. We change the term of "Total SWE" to "snow mass" in whole manuscript
- 10
- 11 Please see below the detailed responses (in blue color).
- 12

**13 REVIEWER 1**

In this manuscript, the authors use a support vector regression (SVR) algorithm that they 15 developed in a previous paper to estimate snow depth from passive microwave observations.

In addition to evaluating their estimates of snow depth against values from GlobSnow and ERA-Interim/Land, they also use snow density assumptions to estimate snow water equivalent (SWE) for the Northern Hemisphere. Their major conclusion is that SWE has been declining by \_5 800 km3 a year, or approximately 139 200 km3 over their 24-year study period. The authors say this decline is equivalent to a 12.5% reduction of SWE over the study period, suggesting the initial amount of SWE was 1113 600 km3.

I believe there is a fundamental flaw in how the authors are calculating annual snow 23 accumulation in this manuscript. Their estimate of annual SWE is orders of magni- tude larger 24 than other global datasets suggest. Mudryk et al. (2015) show that the Northern Hemisphere has 25 an average annual snow accumulation of 3500 km3 (see Figure 1a, taken from Figure 3 in that 26 manuscript). Using four commonly used global datasets (ERA-Interim, GLDAS, MERRA2, and 27 VIC), I estimate the long-term-average global snow storage to be ~4000 km3 (see Figure 1b). 28 Even if these global models/reanalyses are underestimating SWE, it is unlikely they are wrong by 29 as much as this manuscript indicates. I believe the authors may be summing daily values of SWE 30 when calculating their annual total SWE, as one would do when calculating annual precipitation 31 from daily precipitation values. However, this is incorrect when working with SWE. Instead, the 32 authors should consider comparing the annual maximum SWE over their period of record. This 33 will not lead to such a dramatic value of SWE decline, but I think it would be interesting to see 34 how their method compares to changes in SWE from GlobSnow, ERA-Interim/Land, and other 35 global data products.

With this mistake, the manuscript is not ready for publication. But if the authors redo their SWE calculations and the following analyses, I would be interested to see the SWE results from their SVR method. Since this error is critical to the main conclusions of the manuscript, I do not include a review of the rest of the paper.

Reference: Mudryk, L. R., Derksen, C., Kushner, P. J., and Brown, R.: Characterization of
Northern Hemisphere Snow Water Equivalent Datasets, 1981–2010, Journal of Climate, 28,
8037-8051.

Response: Thank you very much for your review of our manuscript. We appreciate your positive
comments and very useful suggestions for improving the manuscript. We made modification
according to your suggestion.

1. The analysis indexes were changed. In Fig. 10, we used annual maximum snow mass, annual average snow mass and annual minimum snow mass to analyze the variation characteristic of snow mass over the past 25 years (1992-2016). The average annual maximum snow mass of NHSnow SWE products have quite same magnitude as the analysis datasets provides by the reviewers and Mudryk et al. (2015), which is approximately 4200 km3.

11

14

Figure 10. Interannual variation of annual maximum snow mass (A), annual average snow mass (B) and annual minimum snow mass (C) over the Northern Hemisphere for three period 1992-2016 (black line), 1992-2001 (blue line), and 2002-2016 (red line). Trends estimates were computed from least squares. P is the confidence level for the coefficient estimates; R2 is the goodness of fit coefficient.

16

Subsequently, we mainly revised the description of Paragraph 2 in Section 4.2, the updateddescription as flowing:

The snow mass variation characteristic over the past 25 years were explored by interannual variation (Fig. 10) and intra-annual cycles (not show figure) of snow mass over the Northern Hemisphere . Figure 10 depicts the time series of interannual variation of annual maximum, average and minimum snow mass with respect to 1992–2016 period. The biggest value of annual maximum snow mass occurred in 1998–1999 up to 4875 km3, while the least was 3969 km3 in 2007-2008. The annual maximum snow mass present particularly significant decreasing trends (P  $\leq 0.05$ ) during 1992–2016, at the rate of approximately -19.88 km3 yr.-1 (Fig. 10A). Trend analysis reveals that annual maximum snow mass have a 8% reduction from 1992 to 2016. Note that it present a increase variation trend by about 25.59 km3 yr.-1 (P > 0.05) rate for 1992-2001. In 2 3 contrast, the annual maximum snow mass exhibits a significantly decrease trends (with -34.80 km3 4 yr.-1,  $P \le 0.05$ ) since 2002, which would lead to a extraordinary decrease during 1992–2016. 5 According to the static, the annual maximum snow mass usually appear in February (about 60%) 6 and March (about 40%), and in recent several years this occurred in March become a normal state. 7 We can find that the biggest and the least value of annual average snow mass respectively appear 8 in 1998-1999 (~2370 km3) and 2015-2016 (~1850 km3) in Fig 10B. Likewise, in Fig 10B and 10C 9 the annual average (minimum) snow mass exhibit a significant decrease trend in 1992-2016 period 10 by rate -19.72 km3 yr.-1, P > 0.05 (-2.00 km3 yr.-1, P  $\leq$  0.05) and 2002-2016 period at a rate of  $-30.70 \text{ km}^3 \text{ yr.}^{-1}$ , P > 0.05 (-2.2 km3 yr.-1, P  $\leq$  0.05). For 1992-2016 period, the variation tendency 11 12 of annual average (minimum) snow mass do not pass the significance level test. Moreover, the 13 reduction for the annual average and annual minimum snow mass is 13% and 67%, respectively.

- 14
- 15

2. We changed the original calculation method of snow mass and only using SWE. The revised

- 17 Table 7 show the variation of monthly average snow mass.
- 18 19

Table 7. Variation rate and changes of monthly average snow mass during 1992-2016. The asterisk indicate that the changes are significant at 95% confidence level

| Month     | Variation rate (km 3 /yr.) | % Change in the mean of monthly average snow mass |
|-----------|---------------------------------------|---------------------------------------------------|
| September | -5.96*                                | -63.89%                                           |
| October   | -25.50*                               | -43.99%                                           |
| November  | -36.50*                               | -26.96%                                           |
| December  | -32.66*                               | -5.00%                                            |
| January   | -34.38*                               | -9.53%                                            |
| February  | -30.89*                               | -11.91%                                           |
| March     | 1.90                                  | -4.30%                                            |
| April     | -4.29                                 | -6.46%                                            |
| May       | -11.33*                               | -19.59%                                           |
| June      | -8.01*                                | -64.67%                                           |

We revised the description of Paragraph 3 to flowing statement:

When analyzing long-term variation of monthly average snow mass, ten months (September to 25 June) exhibit significant decreasing apart from March and April (Table 7). The maximum decrease rate was approximately -36.50 km3 yr.-1 (P  $\leq$  0.05) in November while the minimum decrease 26 27 occurred in April at -4.29 km3 yr.-1 (P > 0.05). An increasing trend appears in March with a rate of 28 approximately 1.90 km3 yr.-1 (P > 0.05), however, relatively large decrement in fall and winter are 29 unable to partially be offset by the increment of March. Compared with the fall (September to 30 November) and spring (March to June), the interannual variability of monthly average snow mass 31 significantly decreased in winter (December to February), with average rate of less than -32 km3 32 yr.-1. The reduction of monthly average snow mass in ten month were generated using the average 33 pattern of each month over 1992-2016 as a reference. We found that the reduction of monthly average snow mass fluctuated ranging from -65% to -4% for each month (September to June) over
1992-2016 (Table 7). The largest and smallest reduction were about 64.67% and 4.30%, which
occurred in June and March, respectively. Variation analysis of monthly average snow mass could
offer a powerful evidence for annual average snow mass exhibit a significantly decreasing
tendency (Table 7, Fig. 10B).

,,

3. We changed (Page 19 Lines 12-15) "Similar conclusions also appear in total SWE change 9 analysis. The total SWE shows a 12.5% reduction and the monthly average total SWE is 65.8% for the largest reduction and a 4.2% for least reduction which occur in June and March, 10 11 respectively. The total SWE report well-documented significant decreasing trends (P  $\leq 0.05$ ) 12 during the study period." to "Similar conclusions also appear in snow mass change analysis. The 13 annual maximum, average and minimum snow mass exhibit significantly decrease trends and 14 respectively show a 8%, 13% and 67% reduction. The monthly average snow mass has shown a 15 decreasing trend almost in every month and the reduction range from 64.67% (June) to 4.3% 16 (March). The annual average snow mass report well-documented significant decreasing trends 17  $(\sim 20 \text{ km}^3 \text{ yr.}^{-1}, P < 0.05)$  during the study period." in Section 5.

4. In Abstract "Further analysis were performed across the Northern Hemisphere during 1992-2016, which used snow depth, total snow water equivalent (snow mass) and, snow cover days as indexes. Analysis showed the total snow water equivalent has a significant declining trends (~5794 km3 yr.-1, 12.5% reduction)" were revised to "Further analysis were performed across the Northern Hemisphere during 1992-2016, which used snow depth, snow mass and, snow cover days as indexes. Analysis showed annual average snow mass has a significant declining trends (~19.72 km3 yr.-1, 13% reduction)."

**Spatiotemporal variation of snow depth in the Northern Hemisphere from 1992 to 2016**

Xiongxin Xiao1, 2, Tingjun Zhang1, 4, Xinyue Zhong3, Xiaodong Li1, Yuxing Li1

1Key Laboratory of Western China's Environmental Systems (Ministry of Education), College of Earth and Environment Sciences, Lanzhou University, Lanzhou 730000, China

2School of Remote Sensing and Information Engineering, Wuhan University, Wuhan 430079, China

3Key Laboratory of Remote Sensing of Gansu Province, Cold and Arid Regions Environmental and

Engineering Research Institute, Chinese Academy of Sciences, Lanzhou 730000, China

4University Corporation for Polar Research, Beijing 100875, China.

*Correspondence to*: Tingjun Zhang (tjzhang@lzu.edu.cn)

Abstract: Snow cover is an effective best indicator of climate change due to its effect on regional and 12 global surface energy, water balance, hydrology, climate, and ecosystem function. We developed a long 13 term Northern Hemisphere daily snow depth and snow water equivalent product (NHSnow) by the 14 application of the support vector regression (SVR) snow depth retrieval algorithm to historical passive 15 microwave sensors from 1992 to 2016. The accuracies of the snow depth product were evaluated 16 against observed snow depth at meteorological stations along with the other two snow cover products 17 (GlobSnow and ERA-Interim/Land) across the Northern Hemisphere. The evaluation results showed 18 that NHSnow performs generally well with relatively high accuracy. Further analysis were performed 19 across the Northern Hemisphere during 1992-2016, which used snow depth, snow mass and, snow 20 cover days as indexes. Analysis showed annual average the snow mass has a significant declining 21 trends (~19.72 km3 yr.-1, 13% reduction) <del>(~5794 km3 yr.+1, 12.5% reduction)</del>. 
[revised manuscript text omitted]
 (maximum, and minimum) in one snow cover year (excluded July and August) were 22 calculated as analysis indexes and also monthly average snow mass in 25 years, which is the sum of 23 daily (or the mean of monthly) total SWE in one snow cover year (or each month of 25 years).

The snow mass variation characteristic over the past 25 years were explored by iInterannual variation (Fig. 10) and intra-annual cycles (not show figure) of total SWEsnow mass over the Northern Hemisphere were used to analyze total SWE variation characteristic over the past 25 years (1992–2016). Figure 108 depicts the time series of interannual variation of annual total SWEmaximum, average and minimum snow mass anomaly-with respect to 1992–2016 reference period. The biggest value of annual maximum snow mass anomaly-occurred in 1998–1999 up to 4875

km3-period, with while the least minimum was 3969 km3 in during 2007-2008<del>2015 2016</del>. It The 2 annual maximum snow mass present particularly significant decreasing trends ( $P \le 0.05$ ) during 1992–2016, at the rate of approximately -5794-19.88 km3 yr.-1 (Fig. 10A). Trend analysis reveals that 3 4 annual maximum total SWEsnow mass have a 812.5% reduction from 1992 to 2016. Note that it There is present a slow increase variation trend rate by about 710-25.59 km3 yr.-1 (P > 0.05) rate for 5 6 1992-2001 period. In contrast, the annual maximum total SWE snow mass exhibits a anomaly 7 significantly decrease trends (with -34.80 km3 yr.-1, P ≤ 0.05) after since 2002-at rate of approximately-8 -9041 km3-yr.+, which may would lead to a extraordinary decreaseing trends of total SWE during 9 1992-2016. According to the static, the annual maximum snow mass usually appear in February 10 (about 60%) and March (about 40%), and in recent several years this occurred in March become a 11 normal state-There was a sudden drop of total SWE in 2008 2009 as found in previous studies . We 12 can find that the biggest and the least value of annual average snow mass respectively appear in 13 1998-1999 (~2370 km3) and 2015-2016 (~1850 km3) in Fig 10B. Likewise, in Fig 10B and 10C the 14 annual average (minimum) snow mass exhibit a significant decrease trend in 1992-2016 period by rate -19.72 km3 yr.-1, P > 0.05 (-2.00 km3 yr.-1,  $P \le 0.05$ ) and 2002-2016 period at a rate of -30.70 km3 15 yr.-1, P > 0.05 (-2.2 km3 yr.-1,  $P \le 0.05$ ). For 1992-2016 period, the variation tendency of annual 16 17 average (minimum) snow mass do not pass the significance level test. Moreover, the reduction for the 18 annual average and annual minimum snow mass is 13% and 67%, respectively. However, oOther 19 factors, for instance, oceanic and atmospheric heat transport, sea ice season wind, and solar insolation 20 anomalies, may have contributed to the fluctuation of total SWEsnow mass (Liu and Key, 2014). 21 Variation of total SWEsnow mass across the Northern Hemisphere could well capture the variation 22 characteristic of the Arctic sea ice extent (Tilling et al., 2015).

When analyzing long-term variation of monthly average total SWEsnow mass, ten months 24 (September to June) exhibit significant decreasing apart from March and April (Table 7). The maximum decrease rate was approximately  $-\frac{1066}{36.50}$  km3 yr.-1 (P  $\leq 0.05$ ) in January-November 25 while the minimum decrease occurred in September April at -4.29177 km3 yr.-1 (P > 0.05). An 26 increasing trend appears in March with a rate of approximately 1.9068 km3 yr.-1 (P > 0.05), however, 27 28 relatively large decrement in fall and winter are unable to partially be offset by the increment of 29 March. Compared with the fall (September to November) and spring (February March to June), the 30 interannual variability of monthly average total SWE snow mass significantly decreased in winter (December to JanuaryFebruary), with average rate of less than -321000 km3 yr.-1. The reduction of monthly average snow mass in ten month were generated using the average pattern of each month over 1992-2016 as a reference. We also found that the reduction of monthly average total SWEsnow mass reduction fluctuated ranging from -6665% to -4% for each month (September to June) over 1992-2016 (Table 7). The largest and smallest reduction were about 65.84.67% and 4.302%, which occurred in June and March, respectively. Variation analysis of monthly average snow mass could offer a powerful evidence for annual average snow mass exhibit a significantly decreasing tendency (Table 7, Fig. 10B).

[revised manuscript text omitted]
 SWEsnow mass change analysis. The total-14 SWE annual maximum, average and minimum snow mass exhibit significantly decrease trends and 15 respectively shows a 8%, 132.5% and 67% reduction. and the monthly average snow mass has shown 16 a decreasing trend almost in every monthm onthly average total SWE and the reduction range from is 17 64.6765.8% (June) tofor the largest reduction and a 4.32% (March)for least reduction which occur in 18 June and March, respectively. The total SWE annual average snow mass report well-documented 19 significant decreasing trends (~20 km3 yr.1, P < 0.05) during the study period. Regression analysis 20 multi-year Northern Hemisphere SCD exhibits a prominent decreasing trend at a rate ranging from 0 to 21 5 day yr.-1. The area of decreasing trends of SCD in EU is much larger than in NA. Unlike the SCD 22 variation rate, its variation level shows that non-significant changes areas dominate the variation 23 pattern across the Northern Hemisphere. An abnormal and interesting phenomenon is that opposite 24 SCD and SD variation trends appear in several regions.

While this study shed light on the spatiotemporal variability trends of snow cover across the Northern Hemisphere using 25-year NHSnow product, we cannot claim NHSnow dataset could completely capture the climate change signal in each region and season. Because of the deficiencies and limitations (e.g. overestimation, underestimation), further efforts should be made to improve the estimation accuracy and robustness of the SD inversion algorithm. Additionally, when more reliable and numerous data become available, we will do more comprehensive validation over higher latitudes and mountainous regions (Dai et al., 2017). Meanwhile, the validation analysis also should be carried out in complex terrain and different land cover types (Tennant et al., 2017;Snauffer et al., 2016). It is recommended that future work focus on the climatic effects and climatological causes in snow cover changes to comprehensively understand the associated snow cover change mechanisms against a climate change background (Huang et al., 2017;Flanner et al., 2011;Cohen et al., 2012).

**6 Acknowledgments**

This study was funded by the National Natural Science Foundation of China (grant nos. 91325202;
41871050; 41801028), National Key Scientific Research Program of China (grant no. 2013CBA01802),
and the Strategic Priority Research Program of Chinese Academy of Sciences (grant nos.
XDA20100103; XDA20100313).

**11 Appendix**

---

## Referee Comment (RC2) · Anonymous Referee #2 · 26 Jun 2019

I reviewed the paper titled, tc-2019-33-AC1-supplement.pdf, which was a revision of the paper after the previous reviewer pointed out a calculation error.

In this study, SWE and snow depth data over North America are developed over a long period of record, 1992-2016, to evaluate spatial and temporal trends in snow mass and snow cover duration during that time. The study uses a SVR method which combines passive microwave data with other variables to estimate snow depth. They compute SWE using seasonally varying density estimates developed for different snow classes. They find overall decreasing trends in snow mass during the study period, particularly after 2002, though results vary regionally and at different rates seasonally.

The paper needs a thorough English language review. The developed data and analysis are interesting and important, but at times it is difficult to understand exactly what

was done. In particular, the variation rate analyses are unclear. I suggest adding equation for this metric, and maybe all of them, to make it obvious what was done. Beyond that, my main feedback is to provide additional high-level details about the SVR method. The way the paper is written it is mandatory that the reader refers to Xiao et al. 2018 in order to understand the process. Enough detail should be given here that the reader has a high-level understanding of the SVR algorithm and the process steps involved.

Specific comments:

Page 6, Line 29: SSM/I is listed twice.

Page 8, line 11: "SVR" hasn't been defined in article yet.

Page 9, lines 15-18: If only 9000 stations are valid, why were 17000 used? How were they selected? Does the map (figure 1) show all the stations, or just the ones used in this study? I would recommend only showing the stations used.

Page 11, line 4: ""SD", which actually is SWE" Can you explain what this means?

Page 11, line 24: List of parameters (DS, A, T, G, L, D) are used in sentence but not defined until later. They should be defined when first used. I would recommend revising the sentence to something like "The snow retrieval process uses various parameters to yield snow depth (Xiao et al. 2018).

Page 13, lines 13-23: I'm not sure what is meant by layers. Do you mean layers within the snowpack? Or are you referring to observations of snow at low, medium and high depths?

Page 13, line 29: the part of the sentence, "or medium-to-deep" doesn't seem like it fits. Should this be removed?

Page 14, equation 2: units of density are wrong if you want SWE in mm. Should use a density ratio (snow density/water density) to keep units consistent. (since density of

water is 1 g/cm3, values will be the same)

Page 14, line 21. Why do you have "(decrease)" here?

Page 16, line 24: I think "shadow" should be "shallow"

Page 17, line 26: Switch the order of "extremely significant decrease" and "significant decrease" so that the 5 grades are listed in order from largest increase to decrease. Same with Figure 8 and Table 5.

Page 18, line 25-26: Can you provide the equation for this metric: "Seasonal average SD was defined as the cumulative SD divided by the days in one snow cover season"? It's not clear to me what is being computed.

Page 20, lines 9-11: Maximum snow mass is occurring later in the year? This is in contrast to most recent literature that is finding max SWE occurring earlier.

Page 20, line 27 (and Table 7): it seems strange that February average snow mass rate decreases significantly, while March increases slightly. Is this snow that accumulated during each month or average snow mass at the time? Can you add text stating why you think that is? It would also be nice to see the data and how the months compare. You could create a figure like Figure 10, showing the time series for all the months in different colors.

---

## Author Comment (AC2) · 14 Jul 2019

Dear editor and reviewer,

Thank you for your positive comments and very important recommendations to improve our manuscript. We have carefully modified the manuscript based on your suggestions and provide a response to each comment. Reviewer comments are given in black, and responses are given in blue. Below we provide a marked-up manuscript version showing the changes based on your comments. The main modifications to the manuscript are as follows:

1. The descriptions about the SVR SD retrieval algorithm were added in *Section 3.2 "Processing*

*flow overview"* according to your suggestions (Page 11)

2. The equations for analysis metric were added in *Appendix* (Page 21)

3. We revised the order "extremely significant increase, significant increase, non-significant change, extremely significant decrease, and significant decrease" to "extremely significant decrease, significant decrease, non-significant change, significant increase, and extremely significant increase" in Table 5, Fig. 8 and Fig. 11 according to your suggestions.

Please see below the detailed responses (in blue color).

**REVIEWER 1#**

I reviewed the paper titled, tc-2019-33-AC1-supplement.pdf, which was a revision of the paper after the previous reviewer pointed out a calculation error.

In this study, SWE and snow depth data over North America are developed over a long period of record, 1992-2016, to evaluate spatial and temporal trends in snow mass and snow cover duration during that time. The study uses a SVR method which combines passive microwave data with other variables to estimate snow depth. They compute SWE using seasonally varying density estimates developed for different snow classes. They find overall decreasing trends in snow mass during the study period, particularly after 2002, though results vary regionally and at different rates seasonally.

The paper needs a thorough English language review. The developed data and analysis are interesting and important, but at times it is difficult to understand exactly what was done. In particular, the variation rate analyses are unclear. I suggest adding equation for this metric, and maybe all of them, to make it obvious what was done. Beyond that, my main feedback is to provide additional high-level details about the SVR method. The way the paper is written it is mandatory that the reader refers to Xiao et al. 2018 in order to understand the process.

Enough detail should be given here that the reader has a high-level understanding of the SVR

algorithm and the process steps involved.

Response: Thank you very much for your review of our manuscript. We appreciate your positive comments and very useful suggestions for improving the manuscript. We made modification according to your suggestion. The point-by-point revisions are as follows.

Specific comments:

Page 6, Line 29: SSM/I is listed twice.

Response: Thanks. We have changed "SSMIS (Special Sensor Microwave Imager), SSM/I (Special Sensor Microwave Imager Sounder)" to "SSM/I (Special Sensor Microwave Imager), SSMIS (Special Sensor Microwave Imager Sounder)" in Page 2, lines 27-28.

Page 8, line 11: "SVR" hasn't been defined in article yet.
Response: We have added the description of SVR, support vector regression in Page 4, line 8.

Page 9, lines 15-18: If only 9000 stations are valid, why were 17000 used? How were they selected? Does the map (figure 1) show all the stations, or just the ones used in this study? I would recommend only showing the stations used.
Response: Thank you very much. This sentence "Data at approximately 30000 meteorological stations were recorded of which 9000 typically are valid" have been revised to "Data at approximately 30000 meteorological stations were recorded of which more than 9000 station are currently obtainable" in Page 5, line 11. Stations with observation dates between 1992 and 2016 were selected in this work. Due to the observation time in some station of this historical dataset are very short, e.g. 1 year, 2 year, even less than one year. Hence, the station used in our work (about 17000) is greater than the number of stations currently obtainable. Figure 1 show the stations finally selected.

Page 11, line 4: "SD", which actually is SWE" Can you explain what this means?
Response: The sentence ""SD", which actually is, is one of the thirteen parameters provided" was changed to "SWE, which is labeled as SD in this dataset, is one of the thirteen parameters provided" in page 6, line 28 .

Page 11, line 24: List of parameters (DS, A, T, G, L, D) are used in sentence but not defined until later. They should be defined when first used. I would recommend revising the sentence to something like "The snow retrieval process uses various parameters to yield snow depth (Xiao et al. 2018).
Response: Than you very much. We changed "The snow retrieval process uses DS and other parameters (A, T, G, L, D ...) to yield snow parameters (e.g. SD, Eq. 1) (Xiao et al., 2018)" to "The snow retrieval process uses various parameters to yield snow parameters (e.g. SD, Eq. 1) (Xiao et al., 2018)" in page 7, lines 18-19

Page 13, lines 13-23: I'm not sure what is meant by layers. Do you mean layers within the snowpack? Or are you referring to observations of snow at low, medium and high depths?
Response: These three layers mean the low, medium and high snow depths. We changed "Firstly, the numbers of sample in the three layers, layer1 ($0 \leq SD < 50$), layer2 ($50 \leq SD < 100$) and layer3 ($SD \geq 100$), should be concretely quantified" to "Firstly, the numbers of sample in the three layers that split up by snow depth should be concretely quantified, i.e. layer1 ($0 \leq SD < 50$; low depth), layer2 ($50 \leq SD < 100$; medium depth ) and layer3 ($SD \geq 100$; high depth)" in page 9, lines 18-20

Page 13, line 29: the part of the sentence, "or medium-to-deep" doesn't seem like it fits. Should this be removed?
Response: Thanks, we removed "or medium-to-depth" in this sentence in page 10, line 1.

Page 14, equation 2: units of density are wrong if you want SWE in mm. Should use a density ratio (snow density/water density) to keep units consistent. (since density of water is 1 g/cm3, values will be the same)

Response: Thanks. We have revised the original formula to " $\text{SWE}(mm) = \text{SD}(cm) \times \rho_{snow}(g/cm^3)/\rho_{water}(g/cm^3) \times 10$ " in page 10, Eq. 2

Page 14, line 21. Why do you have "(decrease)" here?

Response: Thank you. We have removed "(decrease)" in this sentence in page 10 line 23.

Page 16, line 24: I think "shadow" should be "shallow"

Response: We changed "shadow" to "shallow" in page 12 line 23.

Page 17, line 26: Switch the order of "extremely significant decrease" and "significant decrease" so that the 5 grades are listed in order from largest increase to decrease. Same with Figure 8 and Table 5.

Response: Thanks. We have revised the original order to "extremely significant decrease, significant decrease, non-significant change, significant increase, and extremely significant increase" in this sentence (page 13 lines 24-26) and the description in Table 5, Fig. 8 and Fig. 11.

Page 18, line 25-26: Can you provide the equation for this metric: "Seasonal average SD was defined as the cumulative SD divided by the days in one snow cover season"? It's not clear to me what is being computed.

Response: Thanks. This sentences "Seasonal average SD was defined as the cumulative SD divided by the days in one snow cover season" can be described by the following formula.

$$\text{SD}_{average} = \frac{\sum_{i=1}^{n} SD_i}{n} \tag{1}$$

$n$ is the number of days in one snow cover season, $i$ is $i$th day in one snow cover season. This formula have been added in Appendix.

Page 20, lines 9-11: Maximum snow mass is occurring later in the year? This is in contrast to most recent literature that is finding max SWE occurring earlier.

Response: Thank you very much. We have searched and consulted plenty of literature about snow mass or snow water equivalent. Perhaps because of the limited research data available to us, we have not found a study on the occurrence time of maximum snow mass on the hemisphere scale. More research is on the change trend of the snow mass or the maximum SWE. Our finding is consistent with other research conclusions, and it is found that the snow mass or maximum SWE exhibits a decreasing trend in the long-term sequence (Section 4.2). As described in Section 4.2 and Section 3.3, snow mass is calculated by SWE multiplied by snow cover area; SWE is derived from snow depth and snow density. One reason may be due to the temporal and spatial differences in snow depth and snow density distribution, the snow mass finally generated will also vary in different study areas, resulting in different occurrence time for maximum snow mass. Therefore, this conclusions on the hemisphere scale may be contrary to the findings of the small study area or regions. Moreover, with the increase in temperature, precipitation may be another reason that affects the downward trend of the maximum snow mass in March. Kumar et al. (2012) quantified the impacts of more extreme precipitation regimes (MEPR) on the maximum seasonal snow water equivalent and found that MEPR potentially alleviate the maximum seasonal snow water equivalent decrease trend. It should be noted that in our future work, we will further study the impact of extreme precipitation conditions and or climatic factors on the trend of snow mass change in the northern hemisphere.

We added this sentence "This finding need to be further analyzed in the future work by correlation with climatic factors, such as precipitation effects (Kumar et al., 2012)" in page 16 lines 5-6.

Kumar, M., Wang, R., & Link, T.E. (2012). Effects of more extreme precipitation regimes on maximum seasonal snow water equivalent. *Geophysical Research Letters*, 39

Page 20, line 27 (and Table 7): it seems strange that February average snow mass rate decreases significantly, while March increases slightly. Is this snow that accumulated during each month or average snow mass at the time? Can you add text stating why you think that is? It would also be nice to see the data and how the months compare. You could create a figure like Figure 10, showing the time series for all the months in different colors.

Response: Thank you. The statement "An increasing trend appears in March with a rate of approximately 1.9068 km3 yr.-1 (P > 0.05), however, relatively large decrement in fall and winter are unable to partially be offset by the increment of March." have been revised to "However, there are no significant trends in March and April with large interannual variations (Table 7)" in page 16 lines 20-23.

The monthly average snow mass index used in here is the average of the snow mass in each day of this month. So this index only describe snow cover information during this month. Figures 1 exhibits the anomalies of monthly average snow mass (from September to June) from 1992 through 2016 with respect to the 1992–2016 average across the Northern Hemisphere. Table 7 and Figures 1-F showed an significant decrease trends in February average snow mass with $R^2 =0.54$; while Mach average snow mass is no significant trends with large interannual variation, $R^2 =0.009$. In general, monthly average snow mass shows decrease from September to June except March and April, no trends with large interannual variability in March and April.

[Figure]

Figures. 1. The anomalies of monthly average snow mass (from September to June) from 1992 through 2016 with respect to the 1992–2016 average across the Northern Hemisphere. (A) September. (B) October. (C) November. (D) December. (E) January. (F) February. (G) March. (H) April. (I) May. (J) June.

[revised manuscript text omitted]
.  We shortly described the SVR SD retrieval algorithm involved six steps (see Fig. 3): step 1 is data preprocessing about meteorological station SD observations and PM brightness temperature data; Before estimating SD using PM data, it is necessary to identify snow cover and dry snow by a set of criteria in step 2; To segregate the land cover effect on snow cover distribution (step 3) and snow properties evolution effect (step 4), SD retrieval model were established on different land cover types (forest, shrub, prairie, bare-land) and snow cover stages (snow cover accumulation, stabilization and ablation stage); in step 5, we chose SVR as retrieval function (Eq. 1) with specific kernel functions and parameters; step 6 is constructing a set of SD retrieval models trained by the suitable size and quality training samples. The more detailed descriptions of these  steps can refer to the Xiao et al paper (Xiao et al., 2018) not repeated here. Here, we provide more detailed but different descriptions for the SVR SD retrieval algorithm in several steps (cf. Fig. 3).

Step 3. Due to the study period pre-dates MODIS data, we used AVHRR land cover as supplement data. MODIS and AVHRR land cover were reclassified into four classes (forest, prairie, shrub and bare-land) which were bases of constructing SD retrieval sub-model. Table A (in appendix) describes the reclassification scheme of AVHRR land cover is described. MODIS land cover reclassification schemes were documented in Xiao et al. (2018). Because of the relative stability of land cover change, MODIS land cover in 2013 was used for each year during 2013–2016. Similarly, MODIS land cover in 2001 was used in each year during 1998–2001, and AVHRR land cover data were used for 6 years (1992–1997).

Step 6.1 Construction of a subcontinental model. It needs to be stressed that the snow properties in the Eurasia (EU) and North America (NA) exhibit noticed discrepancy especially in snow density. (Zhong et al., 2014;Bilello, 1984). One study pointed out that mean snow density in the former Soviet Union ($0.21 \sim 0.31$ g/cm$^3$) was lower than the data from NA ($0.24 \sim 0.31$ g/cm$^3$) (Bilello, 1984), and also Zhong et al. (2014) explained the possible reasons which resulting in the diversity of snow density in EU and NA. Based on this, we separately constructed the SD retrieval models for EU and NA.

Step 6.2 Training dataset selection is the process of removing redundant features from spatial data. The accuracy of estimated SD primarily depends on training data quality, which also demonstrate the significance of the selection rule of training samples (Xiao et al., 2018). Inputting more data than needed in the training dataset to train SD retrieval model, may lead to overfitting and an estimated SD with high error. In this study, we collected an extremely large number of daily SD records over 25 years, necessitating a optimized selection rule to avoid data information redundancy.

The selection rule proposed in previous research (Xiao et al., 2018) was modified and then it was divided into two steps in here. Firstly, the numbers of sample in the three layers that split up by snow depth should be concretely quantified, i.e. layer1 ($0 \leq SD < 50$; low snow), layer2 ($50 \leq SD < 100$; medium depth) and layer3 ($SD \geq 100$; high depth). 
[revised manuscript text omitted]

The snow mass variation characteristic over the past 25 years were explored by interannual variation (Fig. 10) and intra-annual cycles (not show figure) of snow mass over the Northern

Hemisphere . Figure 10 depicts the time series of interannual variation of annual maximum, average and minimum snow mass with respect to 1992–2016 period. The biggest value of annual maximum snow mass occurred in 1998–1999 up to 4875 km$^3$, while the least was 3969 km$^3$ in 2007-2008. The annual maximum snow mass present particularly significant decreasing trends (P $\leq$ 0.05) during

1992–2016, at the rate of approximately -19.88 km$^3$ yr.$^{-1}$ (Fig. 10A). Trend analysis reveals that annual maximum snow mass have a 8% reduction from 1992 to 2016. Note that it present a increase variation trend by about 25.59 km$^3$ yr.$^{-1}$ (P > 0.05) rate for 1992-2001. In contrast, the annual maximum snow mass exhibits a significantly decrease trends (with -34.80 km$^3$ yr.$^{-1}$, P $\leq$ 0.05) since

2002, which would lead to a extraordinary decrease during 1992–2016. According to the static, the annual maximum snow mass usually appear in February (about 60%) and March (about 40%), and in recent several years this occurred in March become a normal state. This finding need to be further analyzed in the future work by correlation with climatic factors, such as precipitation effects (Kumar et al., 2012). We can find that the biggest and the least value of annual average snow mass respectively appear in 1998-1999 (~2370 km$^3$) and 2015-2016 (~1850 km$^3$) in Fig 10B. Likewise, in

Fig 10B and 10C the annual average (minimum) snow mass exhibit a significant decrease trend in

1992-2016 period by rate -19.72 km$^3$ yr.$^{-1}$, P > 0.05 (-2.00 km$^3$ yr.$^{-1}$, P $\leq$ 0.05) and 2002-2016 period at a rate of -30.70 km$^3$ yr.$^{-1}$, P > 0.05 (-2.2 km$^3$ yr.$^{-1}$, P $\leq$ 0.05). For 1992-2016 period, the variation tendency of annual average (minimum) snow mass do not pass the significance level test. Moreover, the reduction for the annual average and annual minimum snow mass is 13% and 67%, respectively.

Other factors, for instance, oceanic and atmospheric heat transport, sea ice season wind, and solar insolation anomalies, may have contributed to the fluctuation of snow mass (Liu and Key, 2014).

Variation of snow mass across the Northern Hemisphere could well capture the variation characteristic of the Arctic sea ice extent (Tilling et al., 2015).

When analyzing long-term variation of monthly average snow mass (refer to Eq. B in Appendix), ten months (September to June) exhibit significant decreasing apart from March and April (Table 7).

The maximum decrease rate was approximately -36.50 km$^3$ yr.$^{-1}$ (P $\leq$ 0.05) in November while the minimum decrease occurred in April at -4.29 km$^3$ yr.$^{-1}$ (P > 0.05). However, there are no significant trends in March and April with large interannual variations (Table 7).  Compared with the fall (September to November) and spring (March to June), the interannual variability of monthly average snow mass significantly decreased in winter (December to February), with average rate of less than -32 $km^3$ $yr.^{-1}$. The reduction of monthly average snow mass in ten month were generated using the average pattern of each month over 1992-2016 as a reference. We found that the reduction of monthly average snow mass fluctuated ranging from -65% to -4% for each month (September to June) over 1992-2016 (Table 7). The largest and smallest reduction were about 64.67% and 4.30%, which occurred in June and March, respectively. Variation analysis of monthly average snow mass could offer a powerful evidence for annual average snow mass exhibit a significantly decreasing tendency (Table 7, Fig. 10B).

[revised manuscript text omitted]
 snow mass change analysis. The annual maximum, average and minimum snow mass exhibit significantly decrease trends and respectively show a 8%, 13% and 67% reduction. The monthly average snow mass has shown a decreasing trend almost in every month and the reduction range from 64.67% (June) to 4.3% (March). The annual average snow mass report well-documented significant decreasing trends (~20 km$^3$ yr.$^{-1}$, P < 0.05)

during the study period. Regression analysis multi-year Northern Hemisphere SCD exhibits a prominent decreasing trend at a rate ranging from 0 to 5 day yr.$^{-1}$. The area of decreasing trends of SCD

in EU is much larger than in NA. Unlike the SCD variation rate, its variation level shows that non-significant changes areas dominate the variation pattern across the Northern Hemisphere. An abnormal and interesting phenomenon is that opposite SCD and SD variation trends appear in several regions.

While this study shed light on the spatiotemporal variability trends of snow cover across the

Northern Hemisphere using 25-year NHSnow product, we cannot claim NHSnow dataset could completely capture the climate change signal in each region and season. Because of the deficiencies and limitations (e.g. overestimation, underestimation), further efforts should be made to improve the estimation accuracy and robustness of the SD inversion algorithm. Additionally, when more reliable and numerous data become available, we will do more comprehensive validation over higher latitudes and mountainous regions (Dai et al., 2017). Meanwhile, the validation analysis also should be carried out in complex terrain and different land cover types (Tennant et al., 2017;Snauffer et al., 2016). It is recommended that future work focus on the climatic effects and climatological causes in snow cover changes to comprehensively understand the associated snow cover change mechanisms against a climate change background (Huang et al., 2017;Flanner et al., 2011;Cohen et al., 2012).

**Acknowledgments**

This study was funded by the National Natural Science Foundation of China (grant nos. 91325202; 41871050; 41801028), National Key Scientific Research Program of China (grant no. 2013CBA01802), and the Strategic Priority Research Program of Chinese Academy of Sciences (grant nos. XDA20100103; XDA20100313).

**Appendix**

$$SD_{average} = \frac{\sum_{i=1}^{n} SD_i}{n} \tag{A}$$

$$SM_{average} = \frac{\sum_{i=1}^{n} SM_i}{n} \tag{B}$$

Where $n$ is the number of days in one specific period of time (one month, or snow cover year/season), $i$ is $i$th day in one specific period of time (one month, or snow cover year/season). SD is snow depth. SM is snow mass.

[revised manuscript text omitted]

                          standard deviation

| Season | 1992-2016 (Mean ± 1 Std.) | 1992-2001 (Mean ± 1 Std.) | 2002-2016 (Mean ± 1 Std.) |
|---|---|---|---|
| Fall | -0.08 ± 0.11 | -0.01 ± 0.19 | -0.15 ± 0.22 |
| Winter | -0.11 ± 0.40 | 0.06 ± 0.62 | -0.22 ± 0.75 |
| Spring | -0.04 ± 0.25 | 0.02 ± 0.51 | -0.07 ± 0.41 |
| Year | -0.06 ± 0.20 | 0.02 ± 0.35 | -0.11 ± 0.34 |

Table 7. Variation rate and changes of monthly average snow mass during 1992-2016. The asterisk

                          indicate that the changes are significant at 95% confidence level

| Month | Variation rate (km$^3$/yr.) | % Change in the mean of monthly average snow mass |
|---|---|---|
| September | -5.96* | -63.89% |

| | | |
|---|---|---|
| October | -25.50* | -43.99% |
| November | -36.50* | -26.96% |
| December | -32.66* | -5.00% |
| January | -34.38* | -9.53% |
| February | -30.89* | -11.91% |
| March | 1.90 | -4.30% |
| April | -4.29 | -6.46% |
| May | -11.33* | -19.59% |
| June | -8.01* | -64.67% |

[Figure]

Figure 1. Distribution of Meteorological stations overlaid on ETOPO1 in the Northern Hemisphere.

[Figure]

           Figure 2. Snow Class distribution in the Northern Hemisphere

[Figure]

Figure 3. Process flowchart diagram for developing Northern Hemisphere daily snow depth and snow

               water equivalent data

[Figure]

          Figure 4. Flowchart diagram of the generation of NHSnow products.

[Figure]

Figure 5. Bias of each meteorological station and histogram of biases for three products: a), b)

NHSnow; c), d) GlobSnow, e), f) ERA-Interim/Land. The red dashed line in right column figures are

                      the fitted normal distribution curve

[Figure]

Figure 6. MAE of each meteorological station for three products: a) NHSnow, b) GlobSnow, c)

ERA-Interim/Land.

[Figure]

Figure 7. RMSE of each meteorological station for three products: a) NHSnow, b) GlobSnow, c)

ERA-Interim/Land.

[Figure]

Figure 8. The variation rate pattern of season maximum SD with statistical significances over the

Northern Hemisphere for three snow cover season, fall (a; September to November), winter (b;

December to February), spring (c; March to June) from 1992-2016.

[Figure]

Figure 9. The variation rate pattern of season average SD over the Northern Hemisphere for three snow

    cover season, fall (a, b; September to November), winter (c, d; December to February), spring (e, f;

    March to June) from 1992-2016. Black dots in (a, c, e) indicate that the changes are significant at 95%

    confidence level (CL). The zonal distribution in (b, d, f) are mapped at 0.25 degree resolution in

    latitude. The error bars in (b, d, f) is one times of standard deviation.

[Figure]

Figure 10. Interannual variation of annual maximum snow mass (A), annual average snow mass (B)

and annual minimum snow mass (C) over the Northern Hemisphere for three period 1992-2016 (black line), 1992-2001 (blue line), and 2002-2016 (red line). Trends estimates were computed from least squares. P is the confidence level for the coefficient estimates; $R^2$ is the goodness of fit coefficient.

[Figure]

Figure 11. The variation rate pattern of SCD (a) and their statistical significances (c) over the Northern

Hemisphere from 1992-2016. The zonal distribution in (b) are mapped at 0.25 degree resolution in latitude. The error bars in (b) is one times of standard deviation.

---

## Author Comment (AC3) · 29 Jul 2019

Dear editor and reviewer,

Thank you for your positive comments and very important recommendations to improve our manuscript. We have carefully modified the manuscript based on your suggestions and provide a response to each comment. The paper have been polished by a native English speaker. The following revise are based on two reviewers suggestions. Reviewer comments are given in black, and responses are given in blue. Below we provide a marked-up manuscript version showing the changes based on your comments. The main modifications to the manuscript are as follows:

1. Fig. 10 and Table 7 were revised according to the reviewers suggestions.

2. We revised the description in Abstract, Section 4.2 and, Section 5 accordingly.

3. We change the term of "Total SWE" to "snow mass" in whole manuscript

Please see below the detailed responses (in blue color).

**REVIEWER 1#**

In this manuscript, the authors use a support vector regression (SVR) algorithm that they developed in a previous paper to estimate snow depth from passive microwave observations.

In addition to evaluating their estimates of snow depth against values from GlobSnow and ERA-Interim/Land, they also use snow density assumptions to estimate snow water equivalent (SWE) for the Northern Hemisphere. Their major conclusion is that SWE has been declining by _5 800 km3 a year, or approximately 139 200 km3 over their 24-year study period. The authors say this decline is equivalent to a 12.5% reduction of SWE over the study period, suggesting the initial amount of SWE was 1113 600 km3.

I believe there is a fundamental flaw in how the authors are calculating annual snow accumulation in this manuscript. Their estimate of annual SWE is orders of magni- tude larger than other global datasets suggest. Mudryk et al. (2015) show that the Northern Hemisphere has an average annual snow accumulation of 3500 km3 (see Figure 1a, taken from Figure 3 in that manuscript). Using four commonly used global datasets (ERA-Interim, GLDAS, MERRA2, and VIC), I estimate the long-term-average global snow storage to be ~4000 km3 (see Figure 1b). Even if these global models/reanalyses are underestimating SWE, it is unlikely they are wrong by as much as this manuscript indicates. I believe the authors may be summing daily values of SWE when calculating their annual total SWE, as one would do when calculating annual precipitation from daily precipitation values. However, this is incorrect when working with SWE. Instead, the authors should consider comparing the annual maximum SWE over their period of record. This will not lead to such a dramatic value of SWE decline, but I think it would be interesting to see how their method compares to changes in SWE from GlobSnow, ERA-Interim/Land, and other global data products.

With this mistake, the manuscript is not ready for publication. But if the authors redo their SWE calculations and the following analyses, I would be interested to see the SWE results from their SVR method. Since this error is critical to the main conclusions of the manuscript, I do not include a review of the rest of the paper.

Reference: Mudryk, L. R., Derksen, C., Kushner, P. J., and Brown, R.: Characterization of Northern Hemisphere Snow Water Equivalent Datasets, 1981–2010, Journal of Climate, 28, 8037-8051.

Response: Thank you very much for your review of our manuscript. We appreciate your positive
comments and very useful suggestions for improving the manuscript. We made modification
according to your suggestion.

1. The analysis indexes were changed. In Fig. 10, we used annual maximum snow mass, annual
average snow mass and annual minimum snow mass to analyze the variation characteristic of snow
mass over the past 25 years (1992-2016). The average annual maximum snow mass of NHSnow
SWE products have quite same magnitude as the analysis datasets provides by the reviewers and
Mudryk et al. (2015).

[Figure]

Figure 10. Interannual variation of annual maximum snow mass (A), annual average snow mass
(B) and annual minimum snow mass (C) over the Northern Hemisphere for three period 1992-
2016 (black line), 1992-2001 (blue line), and 2002-2016 (red line). Trends estimates were
computed from least squares. P is the confidence level for the coefficient estimates; R2 is the
goodness of fit coefficient.
Subsequently, we mainly revised the description of Paragraph 2 in Section 4.2, the updated
description as flowing (from page 16 lines 18 to page 17 line 13):
"
The snow mass variation characteristic over the past 25 years were explored by interannual variation
(Fig. 10) and intra-annual cycles (not show figure) of snow mass over the Northern Hemisphere .
Figure 10 depicts the time series of interannual variation of annual maximum, average and minimum
snow mass with respect to 1992–2016 period. The biggest value of annual maximum snow mass
occurred in 1998–1999 up to 4875 km3, while the least was 3969 km3 in 2007-2008. The annual
maximum snow mass present particularly significant decreasing trends (P ≤ 0.05) during 1992–
2016, at the rate of approximately -19.88 km3 yr.-1 (Fig. 10A). Trend analysis reveals that annual maximum snow mass have a 8% reduction from 1992 to 2016. Note that it present a increase variation trend by about 25.59 km3 yr.-1 (P > 0.05) rate for 1992-2001. In contrast, the annual maximum snow mass exhibits a significantly decrease trends (with -34.80 km3 yr.-1, P ⩽ 0.05) since 2002, which would lead to a extraordinary decrease during 1992 – 2016. According to the static, the annual maximum snow mass usually appear in February (about 60%) and March (about 40%), and in recent several years this occurred in March become a normal state. This finding needs to be further analyzed in the future work by correlation with climatic factors, such as precipitation effects (Kumar et al., 2012). We find that the biggest and the least value of annual average snow mass respectively appear in 1998-1999 (~2370 km3) and 2015-2016 (~1850 km3) in Fig 10B. Likewise, in Fig 10B and 10C the annual average (minimum) snow mass exhibit a significant decrease trend in 1992-2016 period by rate -19.72 km3 yr.-1, P > 0.05 (-2.00 km3 yr.-1, P ⩽ 0.05) and 2002-2016 period at a rate of -30.70 km3 yr.-1, P > 0.05 (-2.2 km3 yr.-1, P ⩽ 0.05). For 1992-2016 period, the variation tendency of annual average (minimum) snow mass do not pass the significance level test. Moreover, the reduction for the annual average and annual minimum snow mass is 13% and 67%, respectively."

2. We changed the original snow mass calculation method. The revised Table 7 show the variation of monthly average snow mass.

Table 7. Variation rate and changes of monthly average snow mass during 1992-2016. The asterisk indicate that the changes are significant at 95% confidence level. The changes was calculated with respect to the average of monthly average snow mass on 25 years.

| Month | Variation rate (km$^3$/yr.) | % The percentage of Changes |
|---|---|---|
| September | -5.96* | -63.89% |
| October | -25.50* | -43.99% |
| November | -36.50* | -26.96% |
| December | -32.66* | -5.00% |
| January | -34.38* | -9.53% |
| February | -30.89* | -11.91% |
| March | 1.90 | -4.30% |
| April | -4.29 | -6.46% |
| May | -11.33* | -19.59% |
| June | -8.01* | -64.67% |

We revised the description of Paragraph 3 (from page 17 line 18 to page 18 line3) to flowing statement:

"

When analyzing long-term variation of monthly average snow mass (refer to Eq. B in Appendix), ten months (September to June) exhibit significant decreasing apart from March and April (Table 7). The maximum decrease rate was approximately -36.50 km3 yr.-1 (P ⩽ 0.05) in November while the minimum decrease occurred in April at -4.29 km3 yr.-1 (P > 0.05). However, there are no significant trends in March and April with large interannual variations (Table 7).Compared with the fall (September to November) and spring (March to June), the interannual variability of monthly average snow mass significantly decreased in winter (December to February), with average rate of less than -32 km3 yr.-1. The reduction of monthly average snow mass in ten month were generated using the average pattern of each month over 1992-2016 as a reference. We found that the reduction of monthly average snow mass fluctuated ranging from -65% to -4% for each month (September to June) over 1992-2016 (Table 7). The largest and smallest reduction were about 64.67% and 4.30%, which occurred in June and March, respectively. Variation analysis of monthly average snow mass could offer a powerful evidence for annual average snow mass exhibit a significantly decreasing tendency (Table 7, Fig. 10B).
"

3. We changed "Similar conclusions also appear in total SWE change analysis. The total SWE shows a 12.5% reduction and the monthly average total SWE is 65.8% for the largest reduction and a 4.2% for least reduction which occur in June and March, respectively. The total SWE report well-documented significant decreasing trends (P < 0.05) during the study period." to "Similar conclusions also appear in snow mass change analysis. The annual maximum, average and minimum snow mass exhibit significantly decrease trends and respectively show a 8%, 13% and 67% reduction. The monthly average snow mass has shown a decreasing trend almost in every month and the reduction range from 64.67% (June) to 4.3% (March). The annual average snow mass report well-documented significant decreasing trends (~20 km3 yr.-1, P < 0.05) during the study period." in page 21 lines 10-16.

4. In Abstract "Further analysis were performed across the Northern Hemisphere during 1992-2016, which used snow depth, total snow water equivalent (snow mass) and, snow cover days as indexes. Analysis showed the total snow water equivalent has a significant declining trends (~5794 km3 yr.-1, 12.5% reduction)" were revised to "Further analysis were conducted across the Northern Hemisphere during 1992-2016 using snow depth, snow mass and, snow cover days as indexes. Results showed that annual average snow mass had a significant declining trend with a rate of about 19.72 km3 yr.-1 or 13% reduction in snow mass" in page 1 lines 21-25.

**Spatiotemporal variation of snow depth in the Northern Hemisphere from 1992 to 2016**

Xiongxin Xiao[1, 2], Tingjun Zhang[1, 4], Xinyue Zhong[3], Xiaodong Li[1], Yuxing Li[1]

[1]Key Laboratory of Western China's Environmental Systems (Ministry of Education), College of Earth and Environment Sciences, Lanzhou University, Lanzhou 730000, China

[2]School of Remote Sensing and Information Engineering, Wuhan University, Wuhan 430079, China

[3]Key Laboratory of Remote Sensing of Gansu Province, Cold and Arid Regions Environmental and Engineering Research Institute, Chinese Academy of Sciences, Lanzhou 730000, China

[4]University Corporation for Polar Research, Beijing 100875, China.

*Correspondence to*: Tingjun Zhang (tjzhang@lzu.edu.cn)

**Abstract:** Snow cover is an effective  indicator of climate change due to its  impa on regional and global surface energy and  water balance, thus weather and climate, hydrological process and water resources,  and ecosystem as a whole. The overall objective of this study is to investigate changes and variations of snow depth and snow mass over the Northern Hemisphere from 1992 to 2016. We developed a long term Northern Hemisphere daily snow depth and snow water equivalent product (NHSnow) by  applying the support vector regression (SVR) snow depth retrieval algorithm using passive microwave  remote sensing data from 1992 to 2016. Snow depth product were evaluated against observed snow depth at meteorological stations along with the other two snow cover products (GlobSnow and ERA-Interim/Land) across the Northern Hemisphere. The evaluation results showed that NHSnow performs generally well with relatively high accuracy (bias: 0.59 cm, MAE: 15.12 cm and RMSE: 20.11 cm). Further analysis were  conducted across the Northern Hemisphere during 1992-2016 using snow depth, snow mass and  snow cover days as indexes. Results showed that annual average  snow mass had a significant declining trend with a rate of about 19.72 km$^3$ yr.$^{-1}$ or 13% reduction in snow mass . Although spatial variation pattern of snow depth and snow cover days exhibited slight regional differences, it generally reveals a decreasing trend over most of the Northern Hemisphere. Our work provides evidence that rapid changes in snow depth and snow mass are occurring since the beginning  of the 21$^{st}$ century accompanied with dramatic climate warming.

**1. Introduction**

Seasonal snow cover is an important component of the climate system and global water cycle that stores large amounts of freshwater and play major impacts on the surface energy and water budget, thus weather and climate, hydrological processes and water resources, heat and mass exchange between the ground surface and the atmosphere, and ecosystem as a whole (Immerzeel et al., 2010;Zhang, 2005;Robinson and Frei, 2000;Tedesco et al., 2014). On account of the high albedo and low heat conductivity properties of snow, snow cover may directly modulate the land surface energy balance (Flanner et al., 2011), influence on soil thermal regime (Zhang et al., 1996;Zhang, 2005), and indirectly affect atmospheric circulation (Cohen et al., 2012;Zhang et al., 2004;Li et al., 2018). Most jurisdictions in the Northern Hemisphere rely on natural water storage provided by snowpack (Diffenbaugh et al., 2013;Barnett et al., 2005), supplying water for domestic and industrial usage (Sturm, 2015;Qin et al., 2006). Accurate estimation of and reliable information on snow cover spatial and temporal change at regional and global scales is very critical for climate change monitoring, model evaluation and water resources management (Brown and Frei, 2007;Flanner et al., 2011).

Snow depth (SD) is the most useful and commonly measured parameter at national metrological and hydrological stations, numerous research sites, and sites for local and regional water resources assessment programs. Given the sparseness of measurements, it is not possible to fully capture spatial variability of snow cover, especially at high altitude mountains and high latitude regions. Although  in-situ observations can obtain accurate and relative reliable SD and snow water equivalent (SWE) data, it is unrealistic in mountain regions and low population zones because it is labor intensive and high costs. Remote sensing is the most effective and powerful way of obtaining information of snow cover over larger areas (Foster et al., 2011). Optical remote sensing is capable of observing large areas of snow cover; however, it is unable to observe the Earth's surface under cloudy conditions (Foster et al., 2011;Che et al., 2016;Dai et al., 2017). However, microwave remote sensing has this potential and is an attractive alternative to optical remote sensing under all weather conditions and round the clock. It can also be used to estimate SD and SWE due to the interaction with snowpack by providing dual polarization data at different frequencies (Chang et al., 1987;Che et al., 2008;Takala et al., 2011).

[revised manuscript text omitted]

(2009).reported the characteristic of seasonal SCE and snow mass in South America fom to 2006 was described.

   There are, however, very limited data (station data, satellite data or otherwise) that can provide both

SD and SWE on a hemispheric scale. This study describes an approach to develop a consistent

25-year of daily SD and SWE of Northern Hemisphere utilized multi-source data. The primary objective of this study is to develop 25 years (1992-2016) hemispherical SD and SWE products (hereafter referred to as the NHSnow) with a 25 km spatial resolution using support vector regression (SVR) SD retrieval algorithm (Xiao et al., 2018). This paper will address the following questions: 1) How consistent are

NHSnow and other sourced snow cover datasets with  in-situ SD observations? 2) What is the spatiotemporal variability of SD and snow mass in the Northern Hemisphere from 1992-2016?

Meanwhile, it is extremely challenging to make extensive quantitative validation of SD and SWE

estimates.

   This paper is organized in the following five sections. After the introduction section with literature review, the section 2 describes the data sets used in this study. The methods of data preprocessing and snow cover products generation  are provided in Section 3. Next, we describe

NHSnow validation against in-situ snow observations,  demoonstrate the variability of snow cover in the Northern Hemisphere and discuss the potential  controlling factors for the variations of snow cover results utilized NHSnow data (Section 4). Finally, section 5 summarizes the work of this paper.

**2 Datasets**

**2.1 Passive microwave data**

Because cCloud often appears in the snow cover regions or condition, during and the winter season often conceals snowfall possibility which make, here is particularly advantageous for using passive microwave remote sensing to detect SCE and SD. The SSM/I and SSMIS is PM radiometer onboard the United States Defense Meteorological Satellites Program (DMSP) satellite (data available from the National Snow and Ice Data Center, http://nsidc.org/data/NSIDC-0032). The SSM/I (F11 and F13) dataset from this platform, as well as SSMIS (F17), present with the equal-area scale earth grid (EASE-Grid) format and 25 km spatial resolution (Brodzik and Knowles, 2002;Armstrong, 2008;Wentz, 2013;Armstrong and Brodzik, 1995) (Table 1). The snow cover areaSCE and SD derived from SSM/I (F11) and SSM/I (F13) data have high consistency rendering the calibration between these two sensors for snow cover area and SD unnecessary (Dai et al., 2015). To minimize the melt-water effect to some extent, which can change the microwave emissivity of snow, only descending orbit (nighttime) passive microwave data were used (Foster et al., 2009).

**2.2 Ground-based data**

Daily ground based SD measurementsobservation are used to construct and verify the SD retrieval model in this study from two sources. of daily SD observation. The first dataset is the Global Surface Summary of the Day (GSOD) dataset provided by National Oceanic and Atmospheric Administration (NOAA) (https://data.noaa.gov/dataset/dataset/global-surface-summary-of-the-day-gsod). This online dataset, which began in 1929, is derived from the Integrated Surface Hourly (ISH) dataset (Xu et al., 2016). There are fourteen daily elements in GSOD dataset, including SD measured at 0.1 inch. The missing ofSD measurement or reported 0 on the day would be marked 999.9. Data at approximately 30000 meteorological stations were recorded of which more than 9000 typically are typically obtainablevalid. In our study period and area, more than 17 000 meteorological station were selected with records from 1991 to 2016. All meteorological sites and stations are and a location far away from large water bodies such as large rivers, lakes, and oceans.

[revised manuscript text omitted]
. . We shortly describe the SVR SD retrieval algorithm involved six steps (see Fig. 3): step 1 is preprocessing meteorological station SD measurement data and PM brightness temperature data; Before estimating SD using PM data, it is necessary to identify snow cover and dry snow by a set of criteria in step 2; To segregate the land cover effect on snow cover distribution (step 3) and snow properties evolution effect (step 4), SD retrieval model were established on different land cover types (forest, shrub, prairie, bare-land) and snow cover stages (snow cover accumulation, stabilization and ablation stage); in step 5, we chose SVR as retrieval function (Eq. 1) with specific kernel functions and parameters; step 6 is constructing a set of SD retrieval models trained by the suitable size and quality training samples. The more detailed descriptions of these  steps can refer to the Xiao et al  (2018) . Here, we provide more detailed but different descriptions for the SVR SD retrieval algorithm in several steps (Fig. 3)

Step 3. Due to the  study period pre-dates MODIS data, we used AVHRR land cover as supplement data. MODIS and AVHRR land cover were reclassified into four classes (forest, prairie, shrub and bare-land) which were bases of constructing SD retrieval sub-model. Table A (in appendix) describes the reclassification scheme of AVHRR land cover . MODIS land cover reclassification schemes were documented in Xiao et al. (2018). Because of the relative stability of land cover change, MODIS land cover in 2013 was used for each year during 2013–2016. Similarly, MODIS land cover in 2001 was used in each year during 1998–2001, and AVHRR land cover data were used from 1992 through 1997.

Step 6.1 Construction of a subcontinental model. It needs to be stressed that the snow properties in the Eurasia  and North America  exhibit noticed discrepancy especially in snow density. (Zhong et al., 2014;Bilello, 1984). One study pointed out that mean snow density in the former Soviet Union ($0.21 \sim 0.31$ g/cm$^3$) was lower than the data from North America ($0.24 \sim 0.31$ g/cm$^3$) (Bilello, 1984), and also Zhong et al. (2014) explained the possible reasons which resulting in the diversity of snow density in Eurasia and North America. Based on this, we separately constructed the SD retrieval models for Eurasia and North America.

Step 6.2 Training dataset selection is the process of removing redundant features from spatial data. The accuracy of estimated SD primarily depends on training data quality, which also demonstrate the significance of the selection rule of training samples (Xiao et al., 2018). Inputting more data than needed in the training dataset to train SD retrieval model, may lead to overfitting and an estimated SD with high error. In this study, we collected an extremely large number of daily SD records over 25 years, necessitating a optimized selection rule to avoid data information redundancy.

The selection rule proposed in previous research (Xiao et al., 2018) was modified and then it was divided into two steps  here. Firstly, the numbers of sample in the three layers that split up by snow depth should be concretely quantified, i.e. layer1 (0≤SD<50; low snow), layer2 (50≤SD<100; medium depth) and layer3 (SD≥100; high depth), should be concretely quantified. To aviod an inflated training sample in layer2 and layer3, we set a threshold (3 000) determined by several tests (not shown). A threshold (12000) for layer1 was adopted following Xiao et al. (2018). Table 2 summarizes the section of training sample for each layer in detail. After that, the quality of training sample in each layers determined by stratified random sampling is the second step. Stratification was performed in 1 cm SD intervals. Note that, all  select operations  here were randomly performed.

Step 7. Through above steps, the daily estimated SD data in the Northern Hemisphere from January 1992 to December 2016 (excluding July and August) were obtained. Owning to the nature of radiometer observations, NHSnow products are only reliable in areas with seasonal dry snow cover. Areas with sporadic wet or thin snow are not reliably detected and areas marked as snow-free may include areas with wet snow. If one pixel is detected as snow cover by the detection decision tree (Grody and Basist, 1996), but is likely to be shallow  snow with an estimated value of equal or less than 1 cm, the SD value is set as 5 cm (Che et al., 2016;Wang et al., 2008) (Fig. 4.).

Step 8. In this study, Greenland and Iceland are excluded from the generation and analysis of NHSnow (NH_SD, NH_SWE) products due to their complex coastal topography and the difficulty in discriminating snow from ice (Fig. 4) (Brown et al., 2010). Missing data and zero-data gaps occur in the process of generating daily SD gridded products. Therefore, the following filters were applied. Daily estimated SD was averaged with a sliding 7-day window to reduce noise and compensate for missing data in the daily time series. For example, the SD estimate for 4 January is an average of the assimilated scheme output for 1 to 7 January (Takala et al., 2011;Che et al., 2016). When finished, the sliding SD method generated daily SD products for the entire Northern Hemisphere (NH_SD; Fig. 4).

**3.3 Estimation of SWE**

SWE contains more useful information for hydrologists than SD because it represents the amount of liquid water in the snowpack useful for studies on surface hydrological processes and for assessing water resources when  snow melts. One way to estimate SWE is to use SD and snow density ($\rho_{snow}$) 
[revised manuscript text omitted]

The snow mass variation characteristic over the past 25 years were explored by interannual variation (Fig. 10) and intra-annual cycles (not show figure) of snow mass over the Northern Hemisphere . Figure 10 depicts the time series of interannual variation of annual maximum, average and minimum snow mass  with respect to 1992–2016  period. The biggest value of annual maximum snow mass  occurred in 1998–1999 up to 4875 km$^3$ ,  while the least minimum was 3969 km$^3$ in  2007-2008.  The annual maximum snow mass present particularly significant decreasing trends (P ≤ 0.05) during 1992–2016, at the rate of approximately - 19.88 km$^3$ yr.$^{-1}$ (Fig. 10A). Trend analysis reveals that annual maximum snow mass have a 8% reduction from 1992 to 2016. Note that it present a  increase variation trend  by about 7 25.59 km$^3$ yr.$^{-1}$ (P > 0.05) rate for 1992-2001 . In contrast, the annual maximum snow mass exhibits a  significantly decrease trends (with -34.80 km$^3$ yr.$^{-1}$, P ≤ 0.05)

 since 2002 , which  would lead to a extraordinary decreas during 1992–2016. According to the static, the annual maximum snow mass usually appear in February (about 60%) and March (about 40%), and in recent several years this occurred in March become a normal state. This finding needs to be further analyzed in the future work by correlation with climatic factors, such as precipitation effects (Kumar et al., 2012). We find that the biggest and the least value of annual average snow mass respectively appear in 1998-1999 (~2370 km³) and 2015-2016 (~1850 km³) in Fig 10B. Likewise, in Fig 10B and 10C the annual average (minimum) snow mass exhibit a significant decrease trend in 1992-2016 period by rate -19.72 km³ yr.⁻¹, $P > 0.05$ (-2.00 km³ yr.⁻¹, $P \leq 0.05$) and 2002-2016 period at a rate of -30.70 km³ yr.⁻¹, $P > 0.05$ (-2.2 km³ yr.⁻¹, $P \leq 0.05$). For 1992-2016 period, the variation tendency of annual average (minimum) snow mass do not pass the significance level test. Moreover, the reduction for the annual average and annual minimum snow mass is 13% and 67%, respectively. Other factors, for instance, oceanic and atmospheric heat transport, sea ice season wind, and solar insolation anomalies, may have contributed to the fluctuation of snow mass (Liu and Key, 2014). Variation of snow mass across the Northern Hemisphere could well capture the variation characteristic of the Arctic sea ice extent (Tilling et al., 2015).

When analyzing long-term variation of monthly average snow mass (refer to Eq. B in Appendix), ten months (September to June) exhibit significant decreasing apart from March and April (Table 7). The maximum decrease rate was approximately -36.50 km³ yr.⁻¹ ($P \leq 0.05$) in November while the minimum decrease occurred in April at -4.29 km³ yr.⁻¹ ($P > 0.05$). However, there are no significant trends in March and April with large interannual variations (Table 7). Compared with the fall (September to November) and spring (March to June), the interannual variability of monthly average snow mass significantly decreased in winter (December to February), with average rate of less than -1000 km³ yr.⁻¹. The reduction of monthly average snow mass in ten month were generated using the average pattern of each month over 1992-2016 as a reference. We  found that the reduction of monthly average snow mass  fluctuated ranging from -65% to -4% for each month (September to June) over 1992-2016 (Table 7).

The largest and smallest reduction were about 64.67% and 4.30%, which occurred in June and March, respectively. Variation analysis of monthly average snow mass could offer a powerful evidence for annual average snow mass exhibit a significantly decreasing tendency (Table 7, Fig. 10B).

It is very challenging to determine precise distributions of SWE at regional and global scales (Chang et al., 1987;Kongoli, 2004;Tedesco and Narvekar, 2010;Bair et al., 2018). Snow density, which can be used to convert SWE from SD, is  and key factor in accurate estimation of SWE (Sturm et al., 2010;Tedesco and Narvekar, 2010). In fact, snow density typically varies from 0.05 g/cm$^3$ for  fresh snow at low air temperatures to over 0.55 g/cm$^3$ for a ripened snowpack (Anderton et al., 2004;Cordisco et al., 2006). Noteworthily, this study uses dynamic snow density to convert SD to SWE with the assumption that snowpack occurs as a single layer (Sturm et al., 2010). The evolution of the ephemeral snow class was  not  provided by Sturm et al. (2010). The mean value (0.25 g/cm$^3$) of snow density of ephemeral snow (Zhong et al., 2014), which means without any evolution throughout the snow cover year. Meanwhile, Tedesco and Jeyaratnam (2016) used snow density of 0.2275 g/cm$^3$ for ephemeral snow, which is about 10% lower than the value used in this 
[revised manuscript text omitted]
 snow mass change analysis. The annual maximum, average and minimum snow mass exhibit significantly decrease trends and respectively shows a 8%, 13% and 67% reduction. The monthly average snow mass has shown a decreasing trend almost in every month  and the reduction range from 64.67% (June) to 4.3% (March)

. The annual average snow mass report well-documented significant decreasing trends (~20 km$^3$ yr.$^{-1}$, P < 0.05) during the study period.

Regression analysis multi-year Northern Hemisphere SCD exhibits a prominent decreasing trend at a rate ranging from 0 to 5 day yr.$^{-1}$. The area of decreasing trends of SCD in Eurasia is much larger than in North America. Unlike the SCD variation rate, its variation level shows that non-significant changes areas dominate the variation pattern across the Northern Hemisphere. An abnormal and interesting phenomenon is that opposite SCD and SD variation trends appear in several regions.

While this study shed light on the spatiotemporal variability trends of snow cover across the

Northern Hemisphere using 25-year NHSnow product, we cannot claim NHSnow dataset could completely capture the climate change signal in each region and season. Because of the deficiencies and limitations (e.g. overestimation, underestimation), further efforts should be made to improve the estimation accuracy and robustness of the SD inversion algorithm. Additionally, when more reliable and numerous data become available, we will do more comprehensive validation over higher latitudes and mountainous regions (Dai et al., 2017). Meanwhile, the validation analysis also should be carried out in complex terrain and different land cover types (Tennant et al., 2017;Snauffer et al., 2016). It is recommended that future work focus on the climatic effects and climatological causes in snow cover changes to comprehensively understand the associated snow cover change mechanisms against a climate change background (Huang et al., 2017;Flanner et al., 2011;Cohen et al., 2012).

**Acknowledgments**

  This study was funded by the National Natural Science Foundation of China (grant nos. 91325202;

41871050; 41801028), National Key Scientific Research Program of China (grant no. 2013CBA01802), and the Strategic Priority Research Program of Chinese Academy of Sciences (grant nos. XDA20100103;

XDA20100313).

**Appendix**

$$SD_{average} = \frac{\sum_{i=1}^{n} SD_i}{n} \tag{A}$$

$$SM_{average} = \frac{\sum_{i=1}^{n} SM_i}{n} \tag{B}$$

Where $n$ is the number of days in one specific period of time (one month, or snow cover year/season),

$i$ is $i$th day in one specific period of time (one month, or snow cover year/season). SD is snow depth.

SM is snow mass.

[revised manuscript text omitted]

NHSnow; c), d) GlobSnow, e), f) ERA-Interim/Land. The red dashed line in right column figures are

                          the fitted normal distribution curve

[Figure]

Figure 6. MAE of each meteorological station for three products: a) NHSnow, b) GlobSnow, c) ERA-

Interim/Land.

[Figure]

Figure 7. RMSE of each meteorological station for three products: a) NHSnow, b) GlobSnow, c) ERA-Interim/Land.

Figure 8. The variation rate pattern of season maximum SD with statistical significances over the Northern Hemisphere for three snow cover season, fall (a; September to November), winter (b; December to February), spring (c; March to June) from 1992-2016.

[Figure]

Figure 9. The variation rate pattern of season average SD over the Northern Hemisphere for three snow cover season, fall (a, b; September to November), winter (c, d; December to February), spring (e, f;

March to June) from 1992-2016. Black dots in (a, c, e) indicate that the changes are significant at 95%

confidence level (CL). The zonal distribution in (b, d, f) are mapped at 0.25 degree resolution in latitude. The error bars in (b, d, f) is one times of standard deviation.

[Figure]

Figure 10. Interannual variation of annual maximum snow mass (A), annual average snow mass (B)

and annual minimum snow mass (C) over the Northern Hemisphere for three period 1992-2016 (black line), 1992-2001 (blue line), and 2002-2016 (red line). Trends estimates were computed from least squares. P is the confidence level for the coefficient estimates; $R^2$ is the goodness of fit coefficient.

[Figure]

Figure 11. The variation rate pattern of SCD (a) and their statistical significances (c) over the Northern

Hemisphere from 1992-2016. The zonal distribution in (b) are mapped at 0.25 degree resolution in latitude. The error bars in (b) is one times of standard deviation.